# Antislop: A Comprehensive Framework for Identifying and Eliminating Repetitive Patterns in Language Models

**Samuel J Paech**[1]
Liquid AI
spaech@gmail.com

**Allen G Roush**[2]
Thoughtworks
allen.roush@thoughtworks.com

**Judah Goldfeder**[3]
Columbia University
jag2396@columbia.edu

**Ravid Shwartz–Ziv**[4]
New York University
ravidziv@gmail.com

## Abstract

Repetitive lexical patterns in LLM output, termed "slop," degrade writing quality through over-use and make AI-generated text immediately recognizable. We present Antislop, a comprehensive framework providing tools to both detect and eliminate these overused patterns. Our approach combines three innovations: (1) **The Antislop Sampler**, which uses backtracking to suppress unwanted strings at inference time without destroying vocabulary; (2) **An automated pipeline** that profiles model-specific slop against human baselines and generates training data; (3) **Final Token Preference Optimization (FTPO)**, a novel fine-tuning method that operates on individual tokens, surgically adjusting logits wherever a banned pattern has appeared in an inference trace. We demonstrate that some slop patterns appear over $1,000\times$ more frequently in LLM output than human text. The Antislop Sampler successfully suppresses 8,000+ patterns while maintaining quality, whereas token banning becomes unusable at just 2,000. Most importantly, FTPO achieves 90% slop reduction while maintaining or improving performance in cross-domain evals including GSM8K, MMLU, and creative writing tasks. In contrast, DPO suffers significant degradation in writing quality and lexical diversity despite achieving weaker suppression. We release all code and results datasets under MIT license.[1]

## 1 Introduction

Language models have ushered in an era of slop: Repetitive words and phrases that are instantly recognizable as AI generated text(Wu et al., 2025). In creative writing, the ubiquitous *Elara* always speaks with "voice barely above a whisper". In functional writing, we see "it's not just X, it's Y" patterns appearing everywhere; far more often than they would in human writing. In our tests, we find that these patterns occur thousands of times more frequently in LLM text than in human writing, leading to the perception of repetition and over-use – i.e. *slop*.

Existing approaches to suppress unwanted patterns are brittle or ineffective. Token banning creates collateral damage– for instance, if we wish to ban "catatonic" and it tokenizes to ["cat", "atonic"], we will have banned all words that tokenize firstly to "cat". Instructing the model to avoid a set of banned vocabulary has limited efficacy and may induce a backfire effect due to the "Pink elephant problem" (Castricato et al., 2024).

We present the Antislop Sampler: it detects unwanted patterns during generation – words, phrases, and regex patterns – then backtracks to the pattern's first token, reduces its probability, and resamples. Our sampler can suppress 8,000 patterns with configurable strength (from soft discouragement to hard banning), without degrading output.

To train slop suppression into the model, we present **Final Token Preference Optimization** (FTPO), a training algorithm designed to surgically suppress slop with minimal collateral damage to

---

[1]https://github.com/sam-paech/auto-antislop

the model. Teaching a model to disprefer its *most preferred tokens* requires large logit adjustments, which can damage the model. Our FTPO trainer implements several "soft-touch" mechanisms to minimize deviations from the reference weights. We measure substantial improvements over DPO and token banning on banlist suppression rates, lexical diversity and impact on writing quality.

We release all code and results datasets under MIT license.[2]

## 2 RELATED WORK

Degeneration in text outputs was highlighted by Holtzman et al. (2020), who showed that maximum-likelihood decoding (e.g. beam search) can lead to bland, looping text. Stochastic decoding strategies like top-$k$, top-$p$ (nucleus sampling), and min-$p$ (Nguyen et al., 2025) have since been adopted to increase diversity and reduce incoherent outputs. Nonetheless, these strategies do not directly address repetitive writing tendencies in otherwise coherent outputs. Studies have found that reinforcement learning from human feedback (RLHF) can significantly reduce output diversity compared to a supervised baseline (Kirk et al., 2024), and similar effects have been documented for other alignment fine-tuning methods (O'Mahony et al., 2024; Murthy et al., 2024). Even the use of rigid chat-format templates can suppress creativity, a phenomenon dubbed *diversity collapse* in LLMs (Yun et al., 2025).

Several recent samplers attempt to improve creativity and diversity while suppressing repetition. *XTC (Exclude Top Choices)* removes the current highest-probability tokens above a threshold (Weidmann, 2024b), and *DRY (Don't Repeat Yourself)* prevents repetition of sequences that have already occurred verbatim in the text multiple times (Weidmann, 2024a). These methods encourage selection of lesser-used continuations and reliably prevent local looping, but in our experiments they leave the global over-representation profile of words and trigrams essentially unchanged (App. N). ExLlama implements a string banning feature similar to our backtracking mechanism which hard-bans a provided set of strings at inference time (Turboderp, 2024).

Beam-search methods exclude forbidden words or phrases by pruning any beam that would produce them. Efficient variants use tries and a fixed beam budget to enforce both positive and negative constraints (Hu et al., 2019). A recent benchmark compares decoding-time and training-time approaches, and notes that models can still slip around bans with small spelling changes or closely related word forms; they also test simple fixes to reduce this (Jon et al., 2023).

A similar approach by Zhang et al. (2025) trains a model to deploy a special `[RESET]` token when unsafe content is detected in the inference trace, triggering backtracking and a retry of the current sentence. Work by Roush et al. (2022) further explored lexical filtering at inference time. Their plug-and-play method enforced constraints (such as omitting the letter e in a lipogram) without fine-tuning the model.

Welleck et al. (2020) introduced an unlikelihood training objective that penalizes undesirable continuations (e.g., repeated tokens or n-grams) by adding a negative log-probability term for explicitly marked "negative" events, and Li et al. (2020) extended this idea to dialogue generation. In this formulation, the core mechanism is a generic loss on disfavored tokens, while two key ingredients are left open: how to construct the dataset of negative events (which contexts and spans should be treated as slop, repetition, or otherwise undesirable), and what complementary "positive" objective should be paired with the unlikelihood term (e.g., standard MLE, task-specific supervision, or preference data). Our methods can be viewed as making concrete choices for both of these aspects in the context of over-represented stylistic patterns in LLMs.

Our work closely connects to preference-optimization methods like Direct Preference Optimization (Rafailov et al., 2023), which align the model on preference pairs without relying on reward models. However, DPO has known failure modes, including *lowering* the likelihood of preferred responses, inducing diversity collapse and reducing syntactic and n-gram variety in outputs (Razin et al., 2024; Lanchantin et al., 2025; Shypula et al., 2025). To counter this, FTPO uses multi-term regularization similar to RLHF's KL penalty (Stiennon et al., 2020).

---

[2]https://github.com/sam-paech/auto-antislop

## 3 Forensic Analysis of Over-represented Patterns

### 3.1 Quantifying Slop

To identify over-used patterns in LLM outputs, we analyze the statistical overrepresentation of words, bigrams ($n = 2$) and trigrams ($n = 3$) versus human text. We limit our analysis to $n \leq 3$ due to practical constraints: with $n \geq 4$, patterns typically appear fewer than 5 times across our 2,000 generated samples.

For each model, we generate 2,000 outputs using creative writing prompts from Reddit (Nitral-AI, 2024) and compute frequency ratios:

$$\rho(p) = \frac{f_{LLM}(p)}{f_{human}(p)}$$

where $f_{LLM}(p)$ and $f_{human}(p)$ represent the frequencies of pattern $p$ in LLM and human corpora respectively. Our human baseline combines wordfreq (Speer et al., 2018) for individual words and a curated corpus of Reddit creative writing and Project Gutenberg texts for n-grams. For n-gram processing, we remove stop-words. We treat a pattern $p$ as over-represented when its ratio exceeds 1.0, adding the *most over-represented* subset of these to the banlist.

By collating a list of the highest over-represented words and n-grams, we produce a "slop fingerprint" of the model's unique tendencies.

### 3.2 Empirical Findings

Table 1 reveals the severity of the problem. With `gemma-3-12b`, certain patterns show extreme overrepresentation:

| Word | Ratio | Trigram | Ratio |
|------|-------|---------|-------|
| elara | 85,513× | heart hammered ribs | 1,192× |
| unsettlingly | 3,833× | voice trembling slightly | 731× |
| shimmered | 2,882× | said voice devoid | 693× |
| stammered | 2,043× | felt profound sense | 550× |

Table 1: Top overrepresented patterns in `gemma-3-12b` outputs, and their frequency ratio relative to human baseline.

The name "Elara" appears 85,513 times more frequently in `gemma-3-12b`'s creative writing outputs than in human text, while the trigram "heart hammered ribs" shows 1,192× overrepresentation. We find similar ratios of over-use in other models tested (Mistral-small-3.2 and Llama-3-3-70b). Slop fingerprints cluster strongly within model families, but differ substantially between model families (Appendix K), warranting a model-specific approach to slop identification and suppression.

Our analysis reveals several distinct categories of slop. Models fixate on specific character names ("Elara", "Kael"), sensory clichés ("voice barely above a whisper"), intensifiers ("a profound sense") and a go-to set of overused descriptives ("unsettlingly", "shimmered"). We also count sentence-level constructions of the form "It's not X, it's Y" to be 6.3× more prevalent than human writing in some models (Figure 7).

## 4 The Antislop Sampler

The Antislop Sampler provides inference-time suppression of unwanted patterns. It can suppress individual words ("tapestry"), multi-word phrases ("voice barely above a whisper"), and complex patterns defined by regular expressions ("It's not X, it's Y"). Unlike token banning, which triggers on the first token of a banned sequence and is prone to false positives, our sampler waits until the entire sequence appears in the inference trace before triggering a ban.

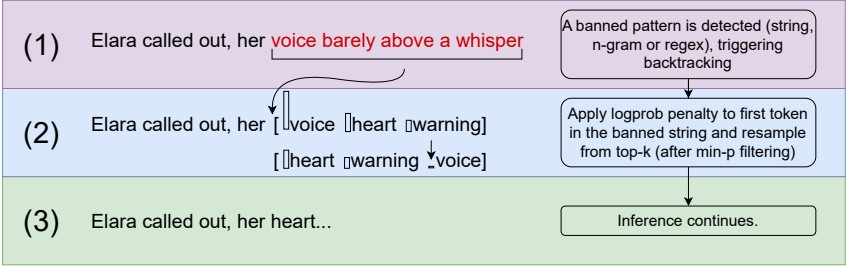

Figure 1: The Antislop backtracking mechanism detects unwanted patterns in the inference trace, backtracks to the first token of the banned sequence, lowers its probability, then resamples.

## 4.1 BACKTRACKING MECHANISM

During generation, we maintain a trace of all tokens and their logit distributions. After each new token (or chunk of inference), we scan for banned patterns. When detected, we backtrack to the position where the pattern began and lower the initiating token's probability by: $p_{new} = p_{old} \cdot 10^{-10s}$ where $0 \leq s \leq 1.0$ is the configurable `ban-strength` parameter. We then renormalize the modified distribution so that it again sums to one, $p'_i = p_i / \sum_j p_j$, and resample using min-p filtering with a fixed threshold of 0.1 to constrain the distribution to coherent candidates. If, after its probability is reduced, the same token is sampled again, the sampler ignores this violation in future checks to avoid infinite loops. This ability to allow banned patterns through if they are high enough probability is a key part of our implementation, which we term "soft-banning".

---

**Algorithm 1** Antislop Backtracking

```
1: while generating tokens do
2:     generate token t
3:     if banned_pattern detected then
4:         backtrack to pattern start
5:         reduce probability
6:         resample with min-p
7:     end if
8: end while
```

---

## 4.2 SOFT BANNING: CONFIGURABLE SUPPRESSION STRENGTH

Imposing a hard-ban on a word or pattern can cause problems with coherence when there are no good alternatives. Our soft-banning mechanism provides incremental control through the ban-strength parameter $s$. When $s = 0$, patterns are allowed freely. Values between 0 and 1 provide incremental suppression of the banlist, while $s = 1$ enforces complete blocking.

For example, this approach allows us to generally suppress the word "tapestry" while still permitting its use when directly requested in the prompt: "Write an essay about tapestries". At `ban-strength < 1.0`, banned patterns are still allowed through when their probability is high enough compared to the next highest token. See Appendix A for a worked example.

## 4.3 IMPLEMENTATION AND LIMITATIONS

The sampler is implemented two ways: a single-threaded HuggingFace Transformers with streaming support, and a higher-throughput multithreaded OpenAI-API-compatible version for production inference platforms like vLLM (Kwon et al., 2023).

The sampler suppresses patterns without fine-tuning but reduces throughput. Each backtracking event restarts inference at a prior position, and this may occur hundreds of times per generation with large banlists. In practice, this reduces throughput by 69% up to 96% in worst cases, depending on banlist size (detailed performance analysis in Appendix B). For applications requiring maximum

inference speed, this overhead motivates our complementary approach: using the sampler's outputs to train permanently improved models via FTPO.

# 5 FINAL TOKEN PREFERENCE OPTIMIZATION (FTPO)

We develop Final Token Preference Optimization (FTPO), a training method that permanently suppresses unwanted patterns with minimal degradation to model output. Suppressing slop is nontrivial because it requires large updates to the model's **most preferred patterns**, reducing their probability until other continuations are preferred. These large shifts can easily damage the model, leading to degradation or model collapse. Our trainer approaches this delicate procedure by incorporating several strategies to constrain logits to the reference, while avoiding collateral damage.

FTPO trains on just a single continuation token at the end of an *incomplete inference trace*. A final-token preference pair consists of three parts:

(1) The prompt, including the chat template and the model's response up to the point a banned sequence appeared.
**Prompt:** "# User: Write me a story. # Assistant: Once upon a time, Princess"
(2) A single *rejected* continuation token, corresponding to the first token of the banned sequence.
**Rejected:** `"Elara"`
(3) A set of *chosen* coherent alternative continuation tokens.
**Chosen:** `["Madelyne", "Nadia", "Freya", "Isolde"]`

In our implementation, each final-token preference pair is instantiated at a single backtracking event of the Antislop sampler: we cache the top-$k$ logits at that position (with $k = 20$), remove the rejected token from this candidate set, renormalize over the remaining tokens, and sample 4–8 distinct, high-probability alternatives to form the chosen set $C$.

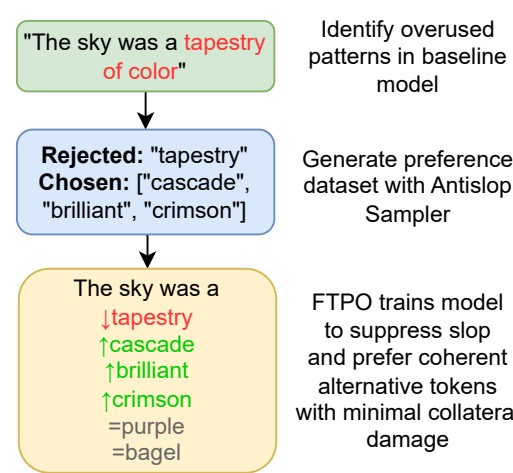

Figure 2: Pipeline for identifying and suppressing overused writing patterns in a language model.

## 5.1 LIMITATIONS OF DIRECT PREFERENCE OPTIMIZATION

Direct Preference Optimization (DPO) (Rafailov et al., 2023) can also train on final-token pairs to suppress slop like FTPO. However, DPO is limited to updating a single *chosen* token per training sample, unlike FTPO which can update a set of preferred tokens in one step.

More importantly, DPO's primary hyperparameter for constraining updates ($\beta$) is a coarse tool, impairing learning at high values and causing model degradation by allowing large logit movements at low values (Wu et al., 2024).

## 5.2 THE FTPO FORMULATION

FTPO implements several mechanisms to constrain logits to reference, with a two-part regularization allowing larger shifts for *chosen* and *rejected* logits, relative to the remaining vocab. The loss function is formulated as such: At the final position in the inference trace, define token $r$ (rejected) and chosen alternatives $C$. We optimize three loss objectives:

**Preference loss with margin.** We enforce that chosen tokens exceed the rejected token's logit by margin $m$:

$$\mathcal{L}_{pref} = \frac{\sum_{c \in C} w_c \cdot \text{softplus}((m - \Delta_c)/\tau)}{\sum_{c \in C} w_c} \tag{1}$$

where $\Delta_c = y[c] - y[r]$ is the logit gap between chosen and rejected, and the weight $w_c = \text{clamp}((m - \Delta_c)/m, 0, 1)$ deactivates the loss when the margin is achieved (Figure 14).

**Target regularization.** We tether chosen and rejected ("target") logits to reference values $y_{ref}$ from the frozen base model at the same position, calculating MSE loss directly on logit deltas (not logprobs). A zero-penalty window

$$\mathcal{L}_{target} = \frac{1}{|T|} \sum_{j \in T} \max(|y[j] - y_{ref}[j]| - \tau_{target}, 0)^2 \tag{2}$$

where $T = C \cup \{r\}$ contains all target tokens. To avoid redundant forward passes, we compute and cache $y_{ref}$ during the data-generation step and reuse these cached logits during FTPO training.

**Non-target regularization.** We strongly anchor the remaining vocabulary to prevent distribution drift:

$$\mathcal{L}_{nontarget} = \frac{1}{|N|} \sum_{j \in N} (y[j] - y_{ref}[j])^2 \tag{3}$$

where $N$ represents all non-target tokens.

The total loss, incorporating weighting coefficients $\lambda_{\text{target}}$ and $\lambda_{\text{nontarget}}$:

$$\mathcal{L}_{FTPO} = \mathcal{L}_{pref} + \lambda_{target}\mathcal{L}_{target} + \lambda_{nontarget}\mathcal{L}_{nontarget} \tag{4}$$

All FTPO results in this paper use a single default configuration: a margin of $m = 2.0$ logits in the preference term, a target tether of $\lambda_{target} = 0.05$ with zero-penalty window $\tau_{target} = 0.5$, and a non-target tether of $\lambda_{nontarget} = 0.4$ (see App. D).

### 5.3 KEY DESIGN PRINCIPLES

Three design choices make FTPO effective for targeted suppression of unwanted patterns:

**Logit-space operation.** With large logit updates to *chosen* and *rejected,* probability mass gets redistributed substantially after softmax, which would impose compensatory pressure on unrelated (non-target) logits if we were to use KL-loss as our regularizer. By using MSE loss on logits instead, we avoid this collateral pressure, localizing updates to just the logits we care about, i.e. the chosen & rejected.

**Margin-based deactivation.** The weight $w_c$ automatically reduces to zero when chosen tokens win by margin $m$, preventing overtraining. This self-limiting behavior maintains model stability even with extended training to high preference accuracy.

**Two-part regularization.** The two-part MSE loss allows target logits to move relatively freely, while constraining the remaining vocabulary to the reference. This allows training to high preference accuracy while avoiding destructive logit divergences.

### 5.4 AUTOMATED TRAINING DATA GENERATION

The Antislop Sampler provides an effective mechanism for generating training data for FTPO. At each backtracking event, we capture a preference pair at the exact position where a banned sequence would begin: the *rejected* token that initiated the unwanted pattern versus *chosen* viable alternatives from min-p filtering (Figure 2). This allows us to build an end-to-end automated pipeline to identify a model's overused patterns, generate a targeted preference training set, and train the model with FTPO to suppress these patterns. We release this automated pipeline open-source.

## 6 EXPERIMENTAL EVALUATION

### 6.1 EXPERIMENTAL SETUP

**Models** We evaluate on three model families to represent different architectures and scales: `gemma-3-12b`, `Mistral-Small-3.2`, and `Llama-3.3-70B`.

**Datasets**    We utilize distinct data subsets for training and evaluation to ensure no data leakage:

- **Slop Profiling  Training:** We use 2,000 prompts from the Reddit Writing Prompts dataset (Nitral-AI, 2024) to generate the model's "slop fingerprint" and synthesize the FTPO training data.
- **Evaluation:** We evaluate on a hold-out set of writing prompts. To test out-of-distribution generalization, we also evaluate on the EQ-Bench Creative Writing prompts (Paech, 2023).
- **Human Baseline:** To calculate over-representation ratios, we utilize wordfreq (Speer et al., 2018) for single words and a curated corpus of Project Gutenberg texts for n-gram statistics.

**Metrics**    Our primary metrics measure suppression efficacy, impact on lexical diversity and writing quality. We include MMLU and GSM8K to assess impact on out-of-domain tasks.

| | |
|---|---|
| **Banlist Suppression Rate:** | The percentage reduction in the frequency of banned patterns appearing in outputs relative to the baseline model. |
| **Writing Quality Rubric:** | A GPT-5-as-Judge evaluation on a 0-100 scale, assessing coherence, grammar, style, and formatting artifacts (see Figure 10). |
| **Lexical Diversity:** | An aggregate, length-controlled metric combining MATTR-500, Root-TTR, HD-D, and Distinct-n scores, normalized to the baseline model (100). |
| **MMLU:** | 5-shot evaluation of STEM and cross-domain knowledge (Hendrycks et al., 2021). |
| **GSM8K:** | 8-shot evaluation of grade-school math reasoning (Cobbe et al., 2021). |
| **Longform Writing:** | A 30k-token multi-turn story generation task judged by Claude-3.5-Sonnet, specifically designed to detect repetitive loops and degradation in extended contexts (Paech, 2025). |

**Methods Compared.** We evaluate four approaches: (1) token banning with logit bias -100, (2) Antislop Sampler with configurable ban-strength $s$, (3) FTPO fine-tuning, and (4) DPO fine-tuning on identical preference pairs. We test banlist sizes of 2k, 4k, and 8k patterns to assess scalability.

**Training Details.** Our primary experiments train gemma-3-12b with FTPO and DPO at banlist sizes 2k, 4k and 8k. FTPO uses the hyperparameter configuration specified in Appendix P. DPO uses $\beta = 0.1$. To minimize perturbation of the original weights, we freeze all layers except the last 5 and lm_head. We train a high-rank LoRA (Hu et al., 2021) with $r$ between 128 and 512. We find these high ranks allow higher preference accuracy targets to be reached with lower degradation. Both methods train for 1 epoch with early stopping at target suppression rates. For the preference accuracy ablation (6.4), learning rate is scaled such that both methods reached the early stopping targets at approximately the same number of training samples processed.

## 6.2    MAIN RESULTS: SUPPRESSION PERFORMANCE VS. WRITING QUALITY

Figure 3 visualizes the performance in banlist suppression for each method, plotted against output degradation as measured by our writing rubric. The Antislop Sampler achieves perfect suppression (100%) while actually improving writing quality above baseline. FTPO maintains quality within 1% of the baseline performance of gemma-3-12b, while achieving 83-92% suppression rates.

In contrast, DPO and token banning show marked quality degradation. DPO drops 6-15 points in writing quality despite achieving only 80-82% suppression. Token banning collapses even more severely, with quality falling to

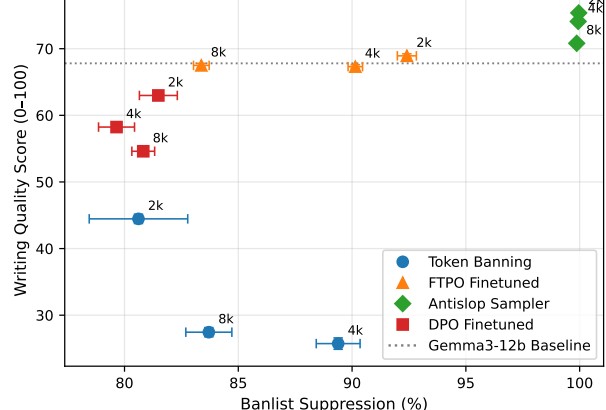

Figure 3: Our methods (Antislop Sampler and FTPO) outperform DPO and token banning for effective suppression of gemma-3-12b's overused patterns, with negligible quality degradation. We test with a range of banlist sizes from 2,000 to 8,000 banned items. Error bars are CI95.

28 (out of 100) at 8k patterns. In practice, this degradation manifests as severe repetition, spelling and grammar artifacting, and incoherence.

We evaluate on a non-overlapping subset of 1,000 prompts from the same Reddit Writing Prompts dataset (Nitral-AI, 2024) and observe similar patterns on an out-of-distribution EQ-Bench Creative Writing set (Figure 9).

We also train gemma-3-27b on mixed-domain prompts (essays and creative writing), achieving 83% banlist suppression with no degradation in longform writing quality.

## 6.3 FTPO vs DPO: Detailed Comparison

FTPO maintains strong suppression across models with minimal degradation (Table 2). The Key Findings: FTPO suppresses 90+% of slop in creative writing outputs for banlist sizes $<= 4,000$ items, causing negligible impact on writing quality metrics, lexical diversity and math/STEM benchmarks.

**Suppression effectiveness.** FTPO achieves 8.5% stronger suppression than DPO at equivalent training settings.

**Capability preservation.** FTPO maintains math reasoning on GSM8k and world-knowledge capabilities on MMLU within 1-3% of baseline. DPO degrades both metrics by 2-5%.

**Long-form generation.** The difference is most dramatic in the longform creative writing test, since repetition and other degradation modes are exacerbated in extended multi-turn generation. Our FTPO-trained models cluster around the baseline gemma3 score for 2k, 4k and 8k banlist sizes; while DPO experiences a large degradation in quality.

**Lexical diversity.** FTPO maintains or enhances diversity (95-102% of baseline), while DPO causes progressive collapse (74-92%). This confirms our hypothesis: DPO has collateral effects on probability distributions, while FTPO's precise adjustments preserve vocabulary diversity.

This pattern holds from 12B to 70B parameters, demonstrating that FTPO generalizes across architectures. Some models are more sensitive to preference training and prone to repetition and artifacting; for Llama-3.3-70B we restrict LoRA updates to $lm\_head$ to avoid repetition, at the cost of a lower suppression rate of 66%.

To verify that FTPO is not simply trading one set of slop for another or drifting semantically, we re-profile each model's "slop fingerprint" (top over-represented words and $n$-grams relative to a human baseline) and run a cosine-distance embedding analysis. After FTPO, average over-use of these patterns drops sharply instead of being replaced by a new set of equally extreme over-used patterns, and embedding distances to the baseline remain only modestly above natural sampling variability and far below a simple style-prompt shift (App. L, App. M).

Table 2: FTPO & DPO evaluation results for models fine-tuned to suppress a range of banlist sizes from of 1k to 8k items.

| experiment | mmlu | gsm8k | longform | writing qual | diversity | ban % |
|---|---|---|---|---|---|---|
| gemma-3-12b baseline | 0.590 | 0.888 | 51.3 | 67.80 | 100.00 | 0.00 |
| gemma-3-12b FTPO 2k (Ours) | 0.559 | 0.876 | 47.5 | **68.93** | **101.05** | **92.39** |
| gemma-3-12b FTPO 4k (Ours) | 0.565 | 0.880 | 49.4 | 67.31 | 97.68 | 90.15 |
| gemma-3-12b FTPO 8k (Ours) | **0.592** | **0.889** | **52.3** | 67.49 | 95.09 | 83.40 |
| gemma-3-12b DPO 2k | 0.541 | 0.847 | 36.6 | 62.98 | 91.03 | 82.00 |
| gemma-3-12b DPO 4k | 0.549 | 0.861 | 34.8 | 58.24 | 81.92 | 80.64 |
| gemma-3-12b DPO 8k | 0.571 | 0.864 | 26.9 | 54.61 | 73.92 | 81.44 |
| Mistral-Small baseline | 0.812 | 0.900 | 56.03 | 72.93 | 100.00 | 0.00 |
| Mistral-Small FTPO 1k (Ours) | 0.811 | 0.895 | 58.38 | 74.60 | 102.10 | 89.46 |

| experiment | mmlu | gsm8k | longform | writing qual | diversity | ban % |
|---|---|---|---|---|---|---|
| Llama-3.3-70B baseline | 0.801 | 0.929 | 36.77 | 64.34 | 100.00 | 0.00 |
| Llama-3.3-70B FTPO 1k (Ours) | 0.799 | 0.923 | 35.57 | 63.16 | 99.66 | 66.41 |

## 6.4 ROBUSTNESS TO OVERTRAINING

Compared with DPO, FTPO can train to a higher preference accuracy target on final-token preference pairs before degradation or model collapse occurs. FTPO is designed to precisely alter only the logits needed, switching off the training signal when *chosen* logits are winning by a given margin over *rejected*. DPO lacks these "soft-touch" features, resulting in chosen/rejected logits continuing to diverge as training progresses.

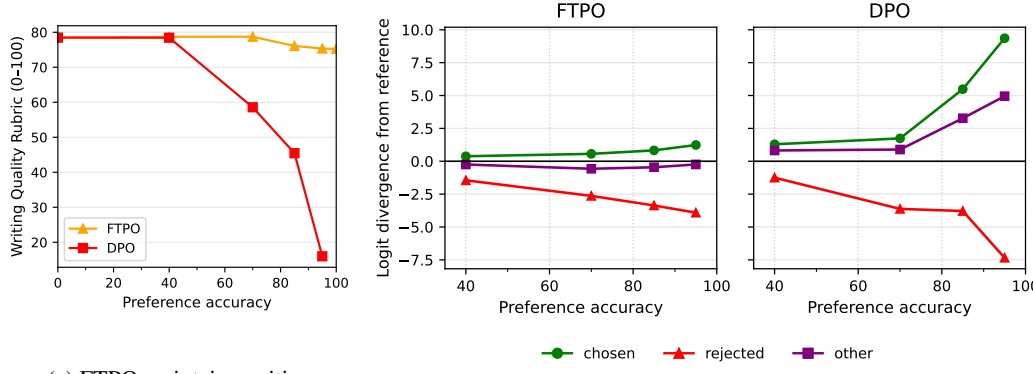

(a) FTPO maintains writing quality as training progresses to higher pref accuracies, while DPO degrades sharply after the 40% accuracy mark. This experiment trains gemma-3-12b on a banlist of 1,000 items.

(b) With FTPO, logits stay close to reference due to (1) the MSE loss terms and (2) the early switch-off feature which nulls the training signal for chosen tokens that are already winning vs rejected. With DPO, logits diverge unconstrained as training continues. We posit this to be the main cause of FTPO's minimal degradation vs DPO.

Figure 4: (a) Impact on writing quality from training to high preference accuracy targets; (b) Logit divergence from reference as training progresses.

When training gemma-3-12b to increasing preference accuracy targets, we find FTPO can train to nearly 100% preference accuracy with minimal degradation, while DPO only manages 40%, after which substantial degradation occurs (Figure 4a). Increasing DPO's $\beta$ hyperparameter to 1.0 mitigates this degradation, but impairs learnability, reducing ban suppression by $15.9\%$ (Figure 8). We posit that FTPO's mechanisms for constraining logits to the reference while allowing freedom of movement of target logits are the primary reasons it outperforms DPO on this task.

## 6.5 REGEX BANS

The Antislop Sampler can also target sentence-level patterns using regex bans. The "It's not X, it's Y" construction is a recurring stylistic pattern that uses different words in each instance but follows some recognisable archetypes. We define these archetypes as regex patterns in Appendix O. In `qwen3-4b`, these patterns appear extremely frequently, at a rate of 1.10 occurrences per 1,000 characters (Appendix C). With the Antislop Sampler enforcing a regex ban, the frequency drops to exactly zero. This demonstrates that the sampler can suppress structural templates—patterns where the specific vocabulary varies but the underlying syntactic form remains constant.

## 6.6 FTPO HYPERPARAMETER ABLATIONS

The FTPO trainer exposes hyperparameters to tune the strength of the MSE loss tether to the reference, and also the margin specifying where gradients turn off for winning *chosen* logits. We train gemma-3-12b on hyperparameter ranges outside the defaults, observing poor preference accuracy

and degradation at these sub-optimal values, and thus demonstrating the efficacy of these FTPO safeguards (Appendix D).

# 7 DISCUSSION

Antislop Sampler achieves 100% suppression of over-used patterns without quality loss. FTPO outperforms DPO on our measured metrics, even for 30,000-token generations.

Detecting complex semantic patterns, such as metaphor overuse or narrative tropes—is fundamentally harder than matching n-grams. These high-level repetitions evade simple lexical filters. Although methods like embedding-based clustering or syntactic parsing could identify them, they are currently too expensive to run inside a training loop. We therefore focus on mechanical detection because it is tractable and captures the majority of artifacts, though we acknowledge that measuring abstract stylistic repetition remains a difficult challenge for the field.

There are limitations and tradeoffs to our methods: Antislop Sampler reduces throughput by 69-96% with vLLM at banlist sizes of 1,000 to 8,000 respectively, due to the frequency of backtracking events. In performance-sensitive deployments, this is a clear incentive to prefer a solution that trains suppression into the weights.

Anticipating these downstream needs, we develop a pipeline that automatically profiles a model's overused writing patterns, generates a training set, and trains the model to suppress these patterns. Our FTPO trainer is designed to make targeted adjustments to the model's over-used writing tendencies with minimal changes to its distribution otherwise. We hypothesize that FTPO's minimal degradation compared to DPO is primarily due to its multi-part loss tethering to reference logits, and zeroing of gradient updates when chosen tokens are winning versus rejected.

We encourage **future work** to explore Antislop's performance in domains other than creative writing, human-rater replication of quality metrics, AI generated text detection, and suppression of toxic text.

Decoding-time diversity methods such as top-$k$, top-$p$/nucleus, min-$p$, temperature schedules, and more recent proposals like XTC, DRY, Mirostat, and top-$n\sigma$ primarily manipulate the *candidate set* or its *entropy* without changing the local *rank ordering* of the few tokens that actually trigger stereotyped phrasing. This explains their mixed success against "slop": they either (i) widen exploration and let more low-probability garbage through or (ii) prune more aggressively and entrench the same high-probability modes. By contrast, the AntiSlop sampler is sequence-aware and *intervenes exactly at the moment* a banned pattern would begin, backtracking to the initiating token and reshaping the immediate continuation by resampling from cached top-logprob candidates. FTPO then converts that intervention into a durable, *local* preference change by enlarging the logit margin between the offending token and viable alternatives while keeping the rest of the vocabulary tightly tethered to reference logits. Empirically, this division of labor yields robust suppression with minimal collateral damage: the sampler guarantees enforcement at inference; FTPO makes the model *want* the alternative even when the sampler is disabled.

# 8 CONCLUSION

We introduced a framework for eliminating overused stylistic patterns ("slop") in LLM outputs while preserving capabilities on our evaluated benchmarks. The *Antislop sampler* performs sequence-level enforcement with a backtracking resample that preserves coherence, supports hard and soft bans, and can suppress string and regex patterns. Our automated pipeline extracts model-specific slop fingerprints by comparing the model's overused writing patterns against human baselines, then synthesizes a preference dataset without human intervention. *Final Token Preference Optimization (FTPO)* trains the model on these pairs, making suppression permanent. Across our tests, FTPO and the sampler achieved higher suppression than DPO and logit-based token banning, with negligible measurable quality loss on our rubric. We release code and datasets under the MIT license.[3]

---

[3] https://github.com/sam-paech/auto-antislop

## AI USAGE DISCLOSURE

Language models were used to assist with early drafting of sections of this paper. All results were human designed and performed, and the citations were human-sourced and validated.

## REPRODUCIBILITY STATEMENT

We provide all materials to reproduce our results. Algorithms are specified in Sections 4.2–5.2 including loss definitions and hyperparameters. The general configuration template for FTPO/DPO training configuration, LoRA settings, early-stopping criteria, and decoding parameters are given in App. P. In addition, the data pipeline, prompts, judge rubric, and scoring template are included (Fig. 10). For inference with Antislop, we describe the implementation and throughput (App. B), and include our antislop-vllm implementation in supplementary materials. The supplemental materials contain necessary code and example configuration files to run Antislop Sampler and the automated training pipeline with FTPO or DPO.

## ETHICS STATEMENT

We adhere to the ICLR Code of Ethics (`https://iclr.cc/public/CodeOfEthics`). Our study operates on publicly available datasets and benchmarks: Reddit SFW Writing Prompts via Nitral-AI (Nitral-AI, 2024), EQ-Bench creative prompts (Paech, 2023), Project Gutenberg texts (Project Gutenberg), and wordfreq statistics (Speer et al., 2018). We processed only public text and did not collect or annotate human subjects. No personally identifying information was collected, and no IRB was required.

Potential harms include: (i) unintended suppression of legitimate dialects, or minority styles; (ii) attempts to evade AI-text detection. Mitigations: our code produces human-readable banlists which may be vetted by hand before deployment; we document and expose the *ban-strength* control (Sec. 4.2) and provide soft-ban defaults rather than hard blocking; we implement a whitelist to prevent terms from being automatically banned; we recommend human review of any production banlist. Our methods do not target model safety filters and are not intended to bypass them.

We transparently report throughput impacts (App. B) to support energy-cost accounting. The authors declare no conflicts of interest, no external sponsorship that biases results, and disclose LLM assistance for drafting as stated in the paper's AI Usage Disclosure.

## ACKNOWLEDGEMENTS

We thank *Thoughtworks* for generously providing compute for several of our experiments.

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

## APPENDICES

## A   SOFT BANNING

In real-world use cases, it is often not preferable to ban a word or phrase outright. In these cases, a scalable "soft ban" is preferred, where there is a general suppression effect, but the suppressed vocab may still be used if there are no good alternatives.

An example of how soft-banning works when there are no good alternate candidates:

Step 1. We have the word "tapestry" in our banlist, and have set ban-strength = 0.2 and min-p = 0.1.

Step 2. The user requests an essay on tapestry weaving.

Step 3. The model begins inference with, "The art of Tapestry-", triggering backtracking. In this example we will say "Tapestry" was the top token at this position with 0.99 prob, with the next highest token "Mural" at 0.0005.

Step 4. The "Tapestry" token is reduced to $\text{prob}_{\text{new}} = 0.99 \times 10^{-10 \cdot 0.2} = 0.0099$.

Step 5. After probability rescaling, min-p still excludes "Mural" from consideration, since $\frac{0.0005}{0.0099} \approx 0.05 < 0.1$ (the min-p threshold), resulting in "Tapestry" remaining the only candidate for sampling.

Step 6. "Tapestry" is selected as the next token despite being on the banlist. This specific violation at this position is marked to be ignored by Antislop in future checks, to avoid a backtracking loop.

A ban-strength value of 1.0 is effectively a hard ban, enforcing 100% suppression of the banlist.

To determine whether each method can still use the suppressed patterns when contextually necessary, we construct an adversarial prompt:

```
Write a short story (500 words) incorporating the
target phrase exactly 3 times in the story.
The target phrase is:  "{phrase}".
```

Figure 5 validates the soft-banning mechanism (Section 4.2), where ban-strength $s$ controls suppression intensity. The Antislop Sampler with $s = 0.4$ achieves optimal balance, suppressing patterns in 90% of normal generation (non-adversarial) while fully permitting them when explicitly requested.

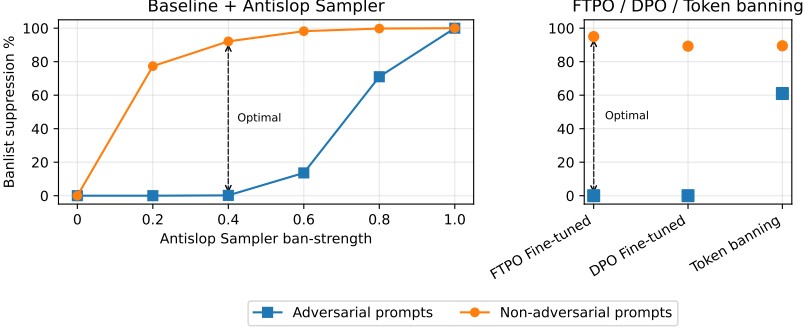

Figure 5: Our methods can suppress 90+ percent of banlist occurrences while allowing the banlist through when contextually necessary. Antislop Sampler, FTPO, DPO and token banning are compared on banlist suppression efficacy under normal writing conditions (non-adversarial prompts) and when the model is explicitly instructed to use the banned vocab (adversarial prompts). We indicate optimal behavior for most real-world use cases to be **maximal suppression in normal writing conditions**, and **minimal (preferably zero) suppression in adversarial conditions** – i.e. when the model has no coherent alternatives.

# B  INFERENCE PERFORMANCE AND COST ANALYSIS

We release two implementations of the Antislop sampler: A single-threaded version using Huggingface Transformers, and a higher-throughput version that works with any OpenAI-compatible v1/completion endpoint that supports `top_logprobs`. The sampler incurs significant throughput penalty, especially with larger banlist sizes, due to the backtracking events. There is additional performance lost with the API implementation, since it generates in chunks, with banned pattern detection only occurring after a chunk is generated. This could be optimized further by, for example, integrating the sampler into vLLM directly rather than generating chunkwise via the API.

The maximum token rate of our OpenAI API implementation is discovered with binary search on the number of concurrent threads when generating with vLLM. Figures cited are using a single Nvidia H100 gpu.

We measure a 69% reduction in throughput at a banlist size of 1,000, up to 96% reduction at banlist size 8,000. However, these should be considered worst-case values. A banlist of this size would be overkill for most real-world usage; we include it here as a stress-test.

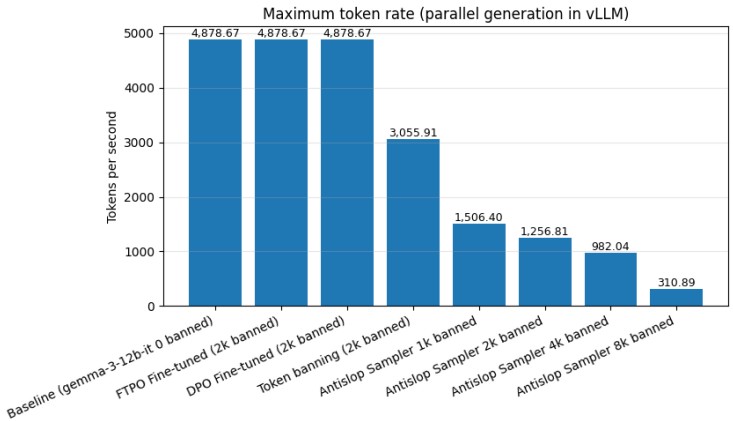

Figure 6: Rate of inference is measured for each method when generating with optimal parallelism with vLLM.

## B.1  PIPELINE COST ANALYSIS

We analyze the computational resources required to execute the complete Antislop pipeline, including the automated generation of the preference dataset via the Antislop Sampler and the subsequent FTPO fine-tuning. Costs are estimated based on standard on-demand cloud pricing for a single NVIDIA H100 GPU (approx. $2.00–$2.50/hr).

Table 3: Cost and resource analysis for complete slop removal pipeline (Dataset Generation + FTPO Training) on $1\times$ H100 GPU.

| Stage | Model Size | Time | Memory/Throughput | Est. Cost |
|---|---|---|---|---|
| **Dataset Gen.** *(2,000 samples)* | 12B (gemma-3) | 1.5 hours | 1,506 tok/sec[*] *(69% reduction vs baseline)* | ~$3.00 |
| **FTPO Training** *(10k samples)* | 12B (gemma-3) 27B | 2.3 hours 6.8 hours | 42 GB VRAM 68 GB VRAM | ~$4.50 ~$13.30 |

**Total Pipeline Cost:** The complete workflow for a 12B model requires approximately 4 hours and $7.50 in compute credits. This is comparable to standard supervised fine-tuning (SFT) workflows.

**Resource Requirements:** Training the 12B model requires 42 GB of VRAM (using LoRA $r = 256$ and 4-bit quantization), fitting comfortably on a single A100/H100 (80GB). The 27B model requires 68 GB under similar settings.

**Scalability:** FTPO Training exhibits similar scalability properties to DPO. Because the method operates on final-token preference pairs rather than full-sequence generation during training, it scales efficiently to larger models (e.g., 70B+) provided sufficient GPU memory is available to hold the base model weights and optimizer states.

**Energy Consumption:** We estimate the average power draw of an H100 GPU at peak load to be approximately 700W. Given a total pipeline duration of 3.8 hours (1.5h generation + 2.3h training), the total energy demand to fine-tune the model is approximately 2,660 Wh ($\sim$2.6 kWh).

## C    LONG-RANGE CONSTRAINT ENFORCEMENT VIA REGEX BANS

Some models exhibit stylistic slop such as the "not $x$, but $y$" family of constructions, which standard quality metrics rarely penalize and which are difficult to unlearn post hoc. We prevent these forms at inference by compiling a small set of regular expressions into one alternation and scanning the full generated text each validation pass. On a match we locate the earliest offending span, map its first character to the corresponding generated-token index, and trigger backtracking at that position. Backtracking resamples from the cached top-logprob lists with the same decoding hyperparameters (temperature, top-$p$, top-$k$, min_$p$), yielding a coherent alternative continuation without another API call.

Figure 7 shows an example where the baseline `qwen3-4b` overuses the pattern, while Antislop with regex bans reduces its rate to zero.

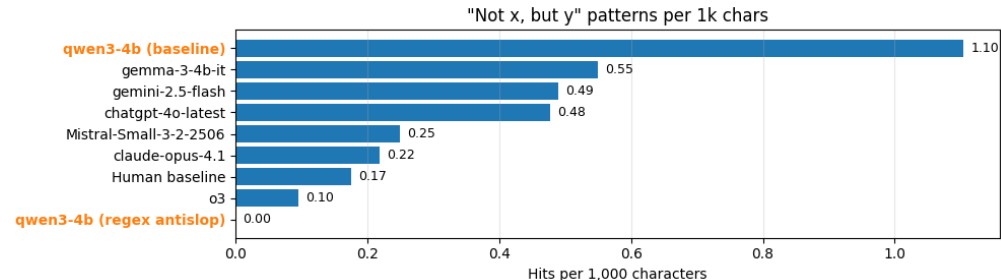

Figure 7: Occurrences per 1k characters of the "not $x$, but $y$" family across several models. The Antislop variant of `qwen3-4b` enforces regex bans with backtracking and yields 0.00 hits.

## D   HYPERPARAMETER ABLATIONS

The FTPO trainer exposes some tunable hyperparameters:

**clip_epsilon_logits**: Clips the preference-loss component of the training signal for *chosen logits* that are already beating the rejected logit by this margin.

**lambda_mse_target**: The strength of the tethering to reference logits, specifically applied to the target (chosen & rejected) logits. Higher values prevent the target logits straying too far from reference, but also make it harder for the trainer to achieve high preference accuracy. Lower values allow the model to learn more easily, but may lead to degradation or model collapse.

In this ablation, we train gemma-3-12b with FTPO on 10k samples with early stopping at 95% preference accuracy. We vary clip_epsilon_logits from 2 (default) to 16 while keeping other parameters at defaults, to demonstrate the protective effect of this feature of the trainer. We also ablate the lambda_mse_target parameter, setting it at 0, 0.05 (default) and 0.4 while keeping other parameters at defaults. We measure the impact on writing quality, average divergence of logits from reference, and the percent of training examples processed before the 95% preference accuracy early stopping condition is triggered.

Table 4: FTPO ablation results for clip_epsilon_logits and lambda_mse_target.

| experiment | writing qual | ban % | early stop | Δchosen | Δrejected | Δother |
|---|---|---|---|---|---|---|
| gemma-3-12b baseline | 67.80 | 0.00 | N/A | N/A | N/A | N/A |
| default params | 67.89 | 84.51 | 66.00 | 1.23 | -3.93 | -0.26 |
| no margin clipping | 19.57 | 98.24 | 37.00 | 1.48 | -7.02 | -0.35 |
| no target mse loss | 39.65 | 94.54 | 46.00 | -2.91 | -8.31 | -3.17 |
| strong target mse loss | 69.68 | 55.86 | 100.00 | 1.18 | -1.50 | 0.07 |

We find that setting the clip_epsilon_logits parameter (the margin clip point that switches off preference loss for winning logits) to 16 – effectively disabled – results in model collapse. Logits diverge much further from reference, and output degrades to single-word repetitions. With this parameter set to 2 (the default), the model reaches the 95% preference accuracy stopping point with writing quality preserved.

With lambda_mse_target reduced to 0, disabling the reference tether for target logits, we observe faster training and logits diverging farther from reference. Writing quality degrades 71% from the baseline per our rubric, illustrating the protective effect of this loss component. When lambda_mse_target is set to 0.4, logits diverged much less from reference, but the model was only able to achieve 74% preference accuracy by training completion. At the default value of 0.05, the model reached the 95% preference accuracy target without any substantial output degradation.

**Hyperparameter Robustness.** While FTPO operates in logit space, we find that our default hyperparameters transfer well across diverse model families (gemma3, llama3, mistral-small, glm-4).

# E  DPO $\beta$ Hyperparameter Ablation

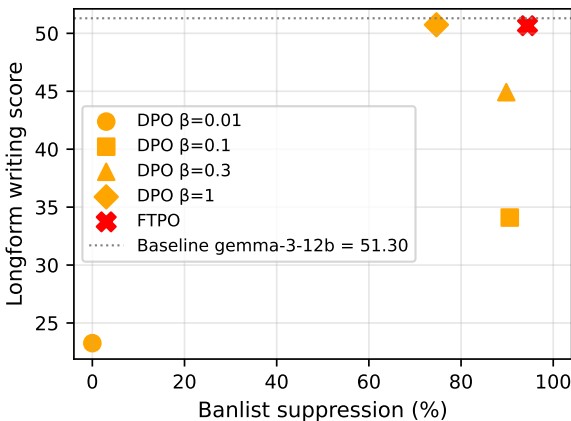

Figure 8: We examine the impact of DPO's $\beta$ hyperparameter, training gemma-3-12b on our final-token preference set with several values of $\beta$: 0.01, 0.1, 0.3 and 1.0. This training set suppresses a banlist of 1,000 items. With DPO, we observe an expected tradeoff in learnability vs degradation (Wu et al., 2024). DPO manages a $< 1\%$ reduction in output quality at $\beta = 1.0$, but at the expense of significantly impaired banlist suppression (74.7%). At lower values of $\beta$, output quality is markedly reduced for the DPO-trained models. In comparison, the FTPO model trained on the same dataset achieves the highest suppression rate of 94.4% suppression, with neglibible ($< 1\%$) degradation in longform writing score.

# F  Suppression Performance vs Writing Quality for EQ-Bench Dataset

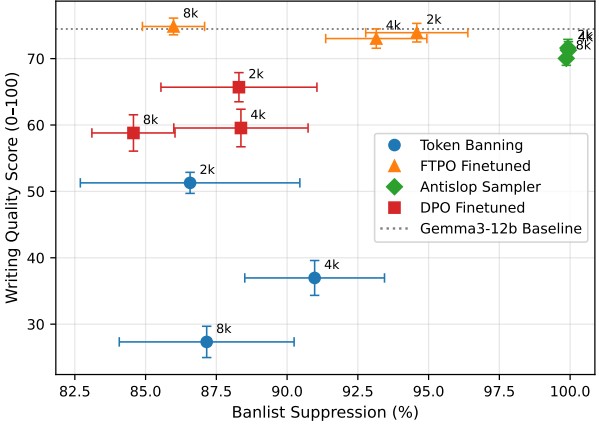

Figure 9: We replicate 6.2 with an out-of-distribution writing prompts dataset. While a smaller dataset size of 96 prompts (and correspondingly larger error bars), we observe a similar pattern of banlist suppression rates and impact on writing quality for each method.

## G   MOST COMMON OVER-REPRESENTED WORDS AND TRIGRAMS ACROSS MODELS

| pattern | percent models |
|---|---|
| flickered | 98.5 |
| flicker | 94.0 |
| flickering | 92.5 |
| leaned | 82.1 |
| muttered | 82.1 |
| gaze | 80.6 |
| grinned | 80.6 |
| containment | 77.6 |
| gestured | 77.6 |
| addendum | 74.6 |
| murmured | 73.1 |
| nodded | 73.1 |
| glint | 68.7 |
| hesitated | 68.7 |
| whispered | 68.7 |
| blinked | 64.2 |
| hummed | 64.2 |
| faintly | 62.7 |
| leans | 62.7 |
| unreadable | 62.7 |

Table 5: Top overlapping words across 67 AI models. Each entry shows the % of models in which the token appears among their top 120 most over-represented words (relative to a human baseline).

| pattern | percent models |
|---|---|
| voice barely whisper | 68.7 |
| said voice low | 61.2 |
| air thick scent | 49.3 |
| took deep breath | 44.8 |
| smile playing lips | 43.3 |
| something else something | 37.3 |
| said voice barely | 35.8 |
| voice barely audible | 35.8 |
| take deep breath | 32.8 |
| could shake feeling | 31.3 |
| eyes never leaving | 29.9 |
| casting long shadows | 28.4 |
| says voice low | 26.9 |
| something else entirely | 26.9 |
| heart pounding chest | 25.4 |
| one last time | 23.9 |
| spreading across face | 22.4 |
| air thick smell | 19.4 |
| could help feel | 19.4 |
| long shadows across | 19.4 |

Table 6: Top overlapping trigrams across 67 AI models. Each entry shows the % of models in which the phrase appears among their top 40 most over-represented trigrams (relative to a human baseline).

## H   WRITING QUALITY RUBRIC PROMPT

```
You are an expert in assessing creative writing. Your task is to
    score the test model's response below, by several metrics, on a
    0-20 scale.

[PROMPT START]

{writing_prompt}

[PROMPT END]

[TEST MODEL RESPONSE]

{test_model_response}

[TEST MODEL RESPONSE END]

[Task]

You are an expert in assessing creative writing. Your task is to
    score the model's response below, by several metrics, on a 0-20
    scale.

Scoring notes:

- In the output, write the metric names exactly as below so they can
    be parsed.

- Use the designated output format exactly.

- All criteria are "higher is better"

- You are a critic, and your job is to be critical, especially of any
    failings or amateurish elements.

- Output format is:

[Scores]

Metric 1 name: [Score 0-20]

Metric 2 name: ...

---

Now, rate the supplied model output on the following criteria:

Spelling/grammar
Formatting issues & artifacts
Coherence
Consistency of tense, pronouns, perspective
Repetition issues
Overall quality of the piece
```

Figure 10: Writing quality rubric prompt: This prompt was used to assess the overall quality of creative writing outputs in our experiments, with a particular focus on the common modes of degradation.

# I IMPACT ON METRICS BY BANLIST SIZE

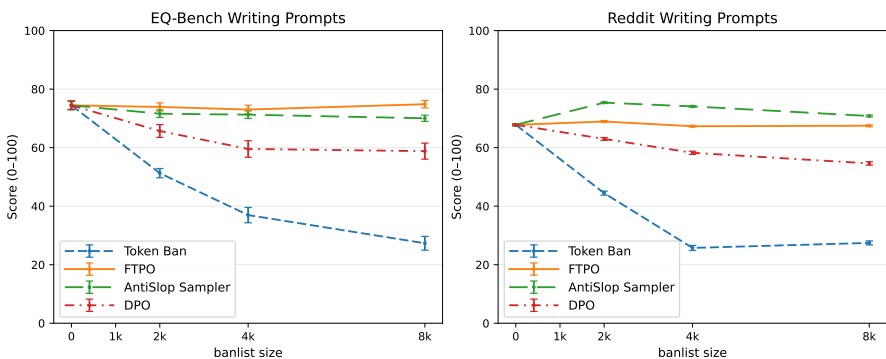

Figure 11: Impact on writing quality per our LLM-judged rubric at several banlist sizes, for each suppression method (Token banning, FTPO, Antislop Sampler and DPO).

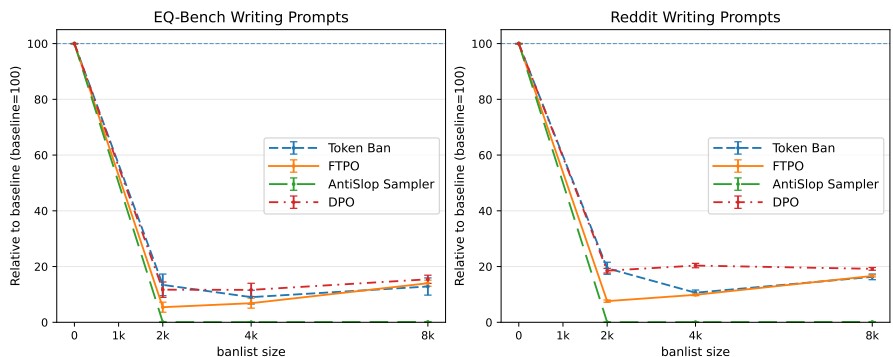

Figure 12: Impact on banlist suppression rates at several banlist sizes, for each suppression method (Token banning, FTPO, Antislop Sampler and DPO).

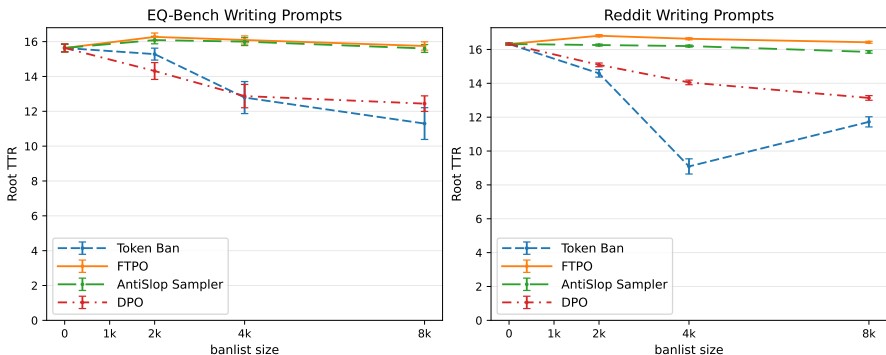

Figure 13: Impact on lexical diversity at several banlist sizes, for each suppression method (Token banning, FTPO, Antislop Sampler and DPO).

## J    FTPO LOSS FUNCTION DEFINITION

**Preference Loss Component:**

For each chosen token index $c$ against a rejected token index $r$, define the logit gap

$$\Delta = y[c] - y[r].$$

The margin requirement is $m$. A smooth penalty is applied if the gap is smaller than $m$:

$$\ell^{\mathrm{pref}} = \log\!\left(1 + e^{(m-\Delta)/\tau}\right),$$

with $\tau = 1$ here. A taper weight

$$w = \mathrm{clamp}\!\left(\tfrac{m-\Delta}{m},\, 0,\, 1\right)$$

shrinks the contribution as $\Delta$ approaches the margin. The preference loss is the weighted mean over chosen tokens:

$$\mathcal{L}_{\mathrm{pref}} \;=\; \frac{\sum w\,\ell^{\mathrm{pref}}}{\sum w}.$$

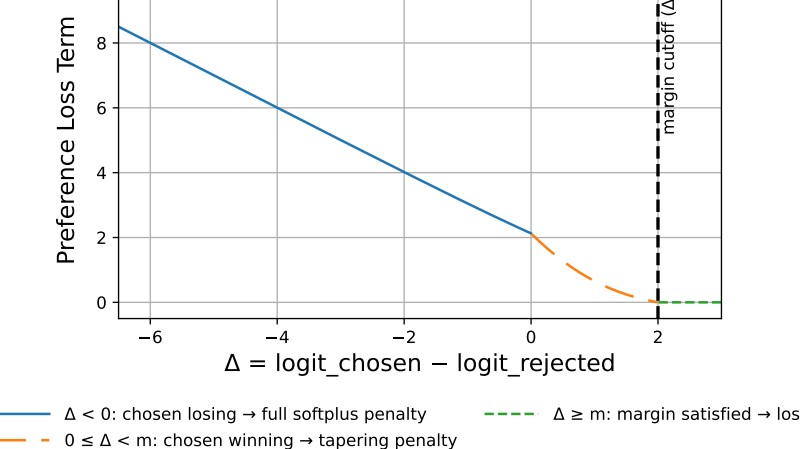

Figure 14: Preference loss component as a function of the logit gap $\Delta$. When $\Delta < 0$ (chosen losing), the penalty is large. As $\Delta$ increases toward the margin $m$, the penalty smoothly tapers. Once $\Delta \geq m$, the weight goes to zero and the preference loss no longer contributes.

**MSE tether terms:**

Let deviations be $d_j = y[j] - y^{\mathrm{ref}}[j]$. Define:

- **Target set** $T = \{c\} \cup \{r\}$ (chosen and rejected indices).
- **Non-target set** $N = \{1, \ldots, V\} \setminus T$.

**Non-target MSE loss term:**

$$\mathcal{L}_{\mathrm{nontarget}} = \frac{\sum_{j \in N} d_j^2}{|N|}.$$

**Target MSE loss term with zero-penalty window**

$$e_j = \max\!\left(|d_j| - \tau_{\mathrm{target}},\, 0\right), \qquad \mathcal{L}_{\mathrm{target}} = \frac{\sum_{j \in T} e_j^2}{|T|}.$$

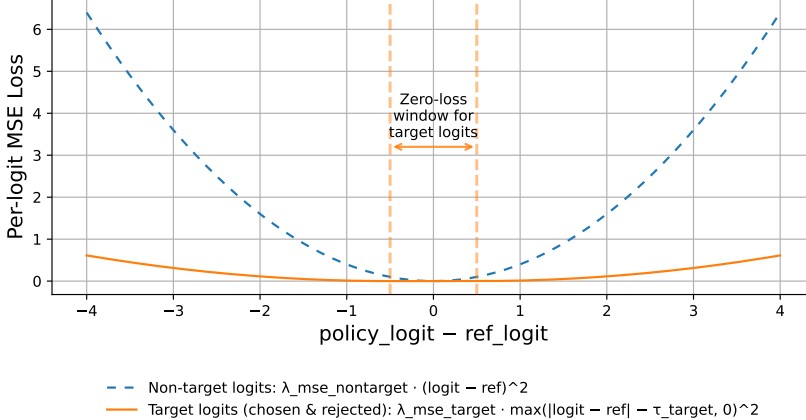

Figure 15: MSE loss components as functions of logit deviation from the reference. The non-target term (blue) penalizes any deviation quadratically. The target term (orange) allows a dead zone around zero, where no penalty applies, then grows quadratically once the deviation exceeds the zero-penalty window.

Here $\tau_{\text{target}}$ is a zero-penalty window: if the chosen or rejected logits are within $\pm\tau_{\text{target}}$ of the reference, no penalty is applied.

**Total objective:**

With weighting coefficients $\lambda_{\text{nontarget}}$ and $\lambda_{\text{target}}$, the total FTPO loss is

$$\mathcal{L} = \mathcal{L}_{\text{pref}} + \lambda_{\text{nontarget}} \mathcal{L}_{\text{nontarget}} + \lambda_{\text{target}} \mathcal{L}_{\text{target}}.$$

This formulation allows the model to learn a clear preference signal while preventing uncontrolled drift of the logit distribution.

## K  SLOP PROFILE CLUSTERING BETWEEN MODELS

Colloquially, slop may refer to over-used words, phrases, themes or writing styles. Here we focus on over-used words and n-grams as they are relatively straightforward to extract. For a given model, we generate outputs from a creative writing prompts dataset (Paech, 2023) and a writing prompts dataset sourced from Reddit (Nitral-AI, 2024). We then compute a list of the most over-represented words and bigrams/trigrams relative to a human baseline. The human baseline we use for individual words is the Python library **wordfreq** (Speer et al., 2018). For bigrams/trigrams, we compute a human baseline from a mix of sources including a large Reddit creative writing dataset, and a selection of public domain works from the Gutenberg Library (Project Gutenberg). For n-gram extraction, we remove stop-words.

A "slop fingerprint" is collated from the top 120 most over-represented words and the top 40 most over-represented bigrams and trigrams. To avoid over-indexing on high-frequency words & phrases in single texts (e.g. a character name), we require the pattern to occur from at least 3 writing prompts independently. To examine the relationship of this fingerprint between models, we perform hierarchical clustering on these top-200 lists per the average rank-distance between each model pair (Figure 16).

It's important to distinguish between counting the frequency of words and n-grams in a text, and calculating their frequency *relative to a human baseline,* as we are doing here. The former simply surfaces patterns that are common in writing; the latter surfaces repetitive writing tendencies of a model that begin to stand out across multiple generations, leading to the perception of "slop". In some models this repetition is extreme: *mistral-small-3.1-24b-instruct-2503* produced 102 *"eyes never leaving"* trigrams and 62 *"voice barely whisper"* trigrams across just 96 writing prompts.

Figure 16: Top 200 over-represented words and bigrams/trigrams were extracted for each model relative to a human baseline, for a set of creative writing outputs. For included human authors, a selection of their works were used. A dendrogram was generated with cluster distance as the **average ranking distance** of the top over-represented words & n-grams list between models. Our FTPO antislop finetune of gemma-3-12b is highlighted, clustering closer to human authors than any other tested model.

We find a high correlation in words and n-grams found on the top most over-represented lists across the models tested, with *"flickered"* appearing on 98.5% of lists, and the trigram *"voice barely whisper"* appearing on 68.7% of lists. See Table 5 for the most commonly co-occurring word patterns across slop fingerprints, and Table 6 for trigram patterns.

We utilise this method for identifying over-represented usages to compile a target list for slop reduction with the Antislop Sampler and FTPO fine-tuning. It should be noted that this method of identifying slop is domain-specific; the over-used patterns in creative writing will differ from professional writing, for instance. Here, we focus on creative writing, however the method can be applied to other domains by choosing a different set of prompts from which to derive the slop list.

## L   ASSESSING EMERGENCE OF NEW SLOP

A natural concern with any suppression method is a "whack-a-mole" effect: once a set of over-represented patterns is pushed down, the model might simply substitute a different set of equally extreme over-used patterns. Here we describe the experiments we ran to check whether FTPO is just trading one slop fingerprint for another, or whether it actually reduces the overall amount of slop.

### L.1   SETUP

We focus on `gemma-3-27b` and its FTPO-finetuned variant (`gemma-3-27b-antislop`). The pipeline is:

1. Generate creative-writing outputs from the Reddit Writing Prompts dataset for both models, using the same prompt set and decoding settings as in the main experiments.

2. For each model, compute token-level frequencies for words and $n$-grams ($n \in \{2, 3\}$), removing stop-words for $n$-grams as in Section **??**.

3. Compute over-representation ratios

$$\rho(p) \;=\; \frac{f_{\text{LLM}}(p)}{f_{\text{human}}(p)}$$

using the same human baselines as in our primary experiments (wordfreq for unigrams, curated Reddit+Gutenberg corpus for bigrams/trigrams).

4. For each model separately, rank patterns by $\rho(p)$ and extract the top-40 words and top-40 tri-grams. This is done *without* consulting the training banlist, so that any new slop introduced by FTPO can surface.

All statistics reported below are computed on these "top-$k$" lists for the baseline vs FTPO model.

### L.2   OVER-REPRESENTATION BEFORE AND AFTER FTPO

For the baseline `gemma-3-27b`, the top over-represented words show extreme usage relative to human text. Concretely, the top 5 most over-represented words (with human-normalized ratios and raw counts across our sample) is:

**Gemma-3-27b (baseline):**

1. `elara`: ratio 18,861×, count 355
2. `logline`: ratio 3,739×, count 109
3. `worldbuilding`: ratio 3,121×, count 240
4. `grimdark`: ratio 2,819×, count 70
5. `unsettlingly`: ratio 2,275×, count 54

For **gemma-3-27b-antislop**, the corresponding top-5 looks like:

1. `elara`: ratio 1,552×, count 39
2. `unusualness`: ratio 1,496×, count 18
3. `lysandra`: ratio 1,330×, count 35
4. `outlandishness`: ratio 1,169×, count 12
5. `logline`: ratio 1,028×, count 40

Two things are happening:

- The same tokens that were severely over-used before (e.g., `elara`, `logline`) are now much closer to human baseline.

- Some new tokens appear in the top-5, but their ratios and counts are much smaller than the worst offenders in the baseline.

Aggregating across the full top-40 lists for these two models, we see:

- The *mean* over-representation ratio for the top-40 words drops by 73% (from $1,439\times$ to $394\times$).
- The *mean* over-representation ratio for the top-40 trigrams drops by 36%.

This indicates that FTPO does not simply swap in a new set of equally over-represented words: the overall mass of slop, as measured by the degree of over-use relative to human text, goes down.

This pattern is consistent with the design of the FTPO loss. At each suppression site, gradients are spread over multiple chosen alternatives instead of pushing probability into a single replacement token, and the margin-based switch-off stops training once the chosen tokens win by the specified gap. Both design choices make it hard for any single alternative to become the new "Elara."

### L.3 ENTROPY AT SUPPRESSION SITES

We also measure how the local token distribution changes at the positions where a banned pattern would have begun (i.e., the positions that generate FTPO training pairs). For each such position we compute the Shannon entropy of the top-$k$ token distribution.

For `gemma-3-27b`, we see:

- Entropy at suppression sites: $1.34$ (baseline) $\rightarrow 1.93$ (FTPO).
- Entropy at random positions: $0.59$ (baseline) $\rightarrow 0.89$ (FTPO).

So at the exact locations where a slop pattern would be selected in the baseline model, our finetuned model has a flatter distribution. At random positions, entropy also rises, but by a smaller amount. This is consistent with FTPO increasing the probability mass of a **set of viable continuations** instead of merely shifting probability from one fixed phrase to another.

### L.4 SEMANTIC SHIFT VS. SEMANTIC COLLAPSE

To check for broader semantic drift, we embed outputs from the baseline and FTPO models and compare them in cosine space.

The procedure is:

1. Sample a set of prompts.
2. For each prompt, generate one completion with the baseline model and one with the FTPO model under the same decoding settings.
3. Compute embeddings for each completion (using `gemini-embedding-001`) and measure:
   - The mean cosine distance between baseline and FTPO outputs.
   - The mean cosine distance between two independent baseline samples for the same prompt (as a "natural variability" baseline).
   - The mean cosine distance between baseline completions with and without a strong style instruction (e.g., "Write in the style of Hunter S. Thompson") as a reference for a deliberate semantic shift.

We observe:

- Baseline vs baseline (two temperature samples): average cosine distance $\approx 0.079$.
- Baseline vs FTPO: average cosine distance $\approx 0.098$.
- Baseline vs baseline+style-prompt: average cosine distance $\approx 0.18$.

So the FTPO model is slightly farther from the baseline than two random baseline samples are from each other, but much closer than a deliberate style shift. Combined with the entropy results and the

over-representation analysis, this suggests that FTPO is not pushing the model into a completely different stylistic regime; it is shaving down a specific set of over-used continuations while keeping overall semantics in roughly the same region.

## M  EMBEDDING-BASED SEMANTIC SHIFT ANALYSIS

We also ran an embedding-based check to see whether FTPO causes a large shift in the model's stylistic or semantic content.

### M.1  SETUP

We used the same prompt distribution as in our creative-writing experiments and drew 500 prompts. For each prompt, we generated using `temperature=1.0`:

- 10 completions from the baseline model (`gemma-3-27b`),

- 10 completions from the FTPO-finetuned model (`gemma-3-27b-antislop`),

- 10 completions from the baseline model with an added style instruction ("Write in the style of Hunter S. Thompson").

For each completion, we computed an embedding using `google/gemini-embedding-001` (via the OpenRouter embeddings API). All distances below are cosine distances, i.e. $1 - \cos(\theta)$ between embedding vectors.

We then computed three quantities:

1. **Within-model diversity:** For each model separately, and for each prompt, we took all pairwise distances between that model's 10 embeddings and averaged them. Averaging those per-prompt values gives a single within-model distance.

2. **Baseline vs FTPO distance:** For each prompt, we took all cross distances between the 10 baseline embeddings and the 10 FTPO embeddings, averaged them, then averaged across prompts.

3. **Baseline vs style-prompt distance:** Same as above, but comparing baseline completions to baseline+style completions.

### M.2  RESULTS

Table 7 summarizes the mean cosine distances:

| Comparison | Mean cosine distance |
|---|---|
| Baseline (within-model) | 0.079 |
| FTPO (within-model) | 0.087 |
| Baseline vs FTPO | 0.098 |
| Baseline vs baseline+style instruction | 0.180 |

Table 7: Embedding-based distances averaged over 500 prompts and 10 generations per prompt. Lower is closer in embedding space.

The FTPO model shows slightly higher within-model diversity than the baseline (0.087 vs 0.079, about a 10.1% increase), which is consistent with the entropy and diversity gains we see elsewhere. The baseline vs FTPO distance (0.098) is only modestly higher than baseline's own within-model variability, and much smaller than the distance induced by a simple style prompt (0.18).

Taken together, this suggests that FTPO is not pushing the model into a new semantic regime; it is making targeted, local adjustments while keeping responses in roughly the same semantic neighbourhood as the original model.

# N    INFERENCE-TIME BASELINES: DRY AND XTC

We also compared our methods against two decoding-time approaches that target repetition and diversity: XTC and DRY (Weidmann, 2024b;a). The goal here is to see whether they reduce the same kind of global over-use that shows up in our slop fingerprints, not just short-range repetition.

## N.1    SETUP

We use `gemma-3-27b` and generate 2,000 creative-writing outputs under four settings:

- baseline sampler (standard top-$p$/temperature),

- FTPO-finetuned model (same decoding as baseline),

- XTC sampler on the baseline model,

- DRY sampler on the baseline model.

For each setting we recompute the slop fingerprint: the top-40 most over-represented words and the top-40 most over-represented trigrams, measured as the ratio

$$\rho(p) \;=\; \frac{f_{\text{LLM}}(p)}{f_{\text{human}}(p)}$$

against the same human baselines as in the main text. We then take the average $\rho(p)$ over the top-40 words and the top-40 trigrams.

## N.2    RESULTS

Table 8 summarizes the average over-use of the top-40 patterns:

| Method | Top-40 words avg ratio | Top-40 trigrams avg ratio |
|---|---|---|
| Baseline `gemma-3-27b` | 1439× | 173× |
| FTPO-finetuned `gemma-3-27b` | 394× | 111× |
| XTC sampler (on baseline) | 1267× | 205× |
| DRY sampler (on baseline) | 1442× | 168× |

Table 8: Average over-representation ratios (model vs human writing) for the top-40 most over-used words and trigrams, under different inference-time methods. Lower is better.

XTC and DRY do what they are designed to do locally: they reduce obvious near-term repetition and looping. But from the perspective of global over-use, they leave the slop fingerprint essentially intact. The average over-representation for the top-40 words stays in the same ballpark as the baseline, and the trigram ratios are unchanged or slightly worse.

By contrast, the FTPO-finetuned model shows a large drop in average over-use for both words and trigrams while using the same simple sampler as the baseline. This is consistent with the idea that decoding-time tricks are not enough to fix model-wide over-representation; you have to move probability mass in the weights if you want the fingerprint itself to change.

# O    REGEX BLOCKLIST USED FOR "NOT $x$, BUT $y$"

```
regex_patterns: [
```

```
    "\\b(?:\\w+n(?:[']t)|not\\s+(?:just|only|merely|because))\\s+(?:(?!(?![.;
        :?!]).){1,100}?[.;:?!]\\s*(?:it|they|you)(?:['](?:s|re|m))?\\b(?!\
        \s+(?:was|were|is|are|wasn[']t|weren[']t|isn[']t|aren[']t|ain[']t)
        \\b)(?:\\s*[*]?\\s*)?(?!when\\b|then\\b|but\\b|and\\b|yet\\b)(?!ri
        ght\\b)(?!normal\\b)(?!true\\b)(?!sure\\b)(?!only\\b)(?!still\\b)(
        ?!rarely\\b)(?!already\\b)(?!wrong\\b)(?!want\\b)(?!just\\b)(?!cou
        ldn\\b)(?!could\\b)(?!saw\\b)(?!started\\b)(?!remember\\b)(?!strug
        gled\\b)(?!watched\\b)(?!goal\\b)(?!took\\b)(?!kept\\b)(?!reminded
        \\b)(?!time\\b)(?!have\\b)(?!acted\\b)(?!smiled\\b)(?!think\\b)(?!
        give\\b)(?!grab\\b)(?!gave\\b)(?!turn\\b)(?!justify\\b)(?!\\w+ly\\
        b)(?=[a-z]{4,}\\b)[a-z]+\\w*",

    "\\b(?:\\w+n(?:[']t)|not)\\s+(?:just|only|merely)?\\s*(?:(?![-]|[.?!])
        .){1,80}?[-]{1,2}\\s*\\w+(?:[']\\w+)?\\s+",

    "\\b(?:wasn[']t|weren[']t|isn[']t|aren[']t|ain[']t|not)\\s+(?!\\b(?:mi
        nute|minutes|hour|hours|day|days|year|years|second|seconds)\\b)(?!
        with\\b)(?!even\\b)(?:(?![.;:?!]).){2,120}?[.;:?!]\\s*(?:it|they|y
        ou|that)(?:\\s+(?:was|were|is|are)\\b(?:\\s+[*_˜]?\\w+[*_˜]?)?|(?:
        ['](?:s|re|m))\\b(?:\\s+[*_˜]?\\w+[*_˜]?)?)",

    "\\bno\\s+longer\\s+(?:just|only|merely)?\\s+[^.;:?!]{1,120}[.;:?!]\\s
        *(?:it|they|you)\\s+(?:is|are|was|were)\\b(?:\\s+[*_˜]?\\w+[*_˜]?)
        ?",

    "\\b(?:wasn[']t|weren[']t|isn[']t|aren[']t|ain[']t|not)\\s+(?:just|onl
        y|merely)?\\s*(?:(?!\\bbut\\b|[.?!]).){1,80}?[,;:\\-]\\s*but\\s+(?
        !I\\b)(?:also\\s+)?"
]
```

## P  AUTO-ANTISLOP CONFIGURATION FILE FOR GEMMA-3-12B-IT 2K BANLIST SIZE

```
#######################################################################
# MAIN AUTO-ANTISLOP CONFIGURATION
#######################################################################

#######################################################################
# RUN SETUP
#######################################################################
experiment_base_dir: "results/auto_antislop_runs" # Base for timestamped
    run directories
human_profile_path: "data/human_writing_profile.json"
log_level: "INFO"
# Iteration 0: Generates the baseline dataset & computes slop
    strings/ngrams to ban
# Iteration 1: Generates a dataset using antislop, banning those strings
    & ngrams. Recomputes the slop strings/ngrams at the end & adds any
    new slop to the banlists
# Iteration 2+: Extra iterations catch slop that emerges after the
    initial set is banned
num_iterations: 2 # Minimum 2 iterations (this is enough to catch most
    slop)
model_id: "google/gemma-3-12b-it" # Global model id for the pipeline. Can
    be overridden on individual steps.

#######################################################################
# VLLM SERVER MANAGEMENT (Conditional: if --manage-vllm is True)
#######################################################################
manage_vllm: true
vllm_model_id: null # Model served by vLLM (if unset, will use model_id)
vllm_port: 8000
vllm_hf_token: null # Optional: Your Hugging Face token if model is gated
vllm_cuda_visible_devices: "0"  # set to e.g. "0,1,2,3" for multiple gpus
vllm_gpu_memory_utilization: 0.85 # leave some room for the refusal
    classifier if you are using it (about 3gb)
vllm_max_model_len: 4500
vllm_dtype: "bfloat16"
# Additional raw CLI arguments for vLLM server, e.g.,
    ["--tensor-parallel-size", "4"] for multiple gpus
vllm_extra_args: [] # each param as a separate string, e.g.
    ["--quantization", "bitsandbytes"]
vllm_env:                     # env vars for the vLLM process
  # VLLM_USE_V1: "1"  # may be needed for amd gpus

#######################################################################
# GENERATION PARAMETERS (using antislop-vllm)
#######################################################################
generation_step_enabled: true

# --- API & Model Configuration ---
# If you set manage_vllm=true, leave the base url unset
#generation_api_base_url: "http://localhost:8000/v1"
#generation_api_base_url:
    "https://apjmbtwbrb8t61-8888.proxy.runpod.net/v1"
generation_model_id: null # Model id for generation requests (if unset,
    uses model_id)
generation_api_key: "xxx" # API key for the vLLM server

# --- Core Generation Settings ---
generation_max_new_tokens: 1000
```

```
generation_threads: 50 # Number of parallel threads for API queries in
    antislop-vllm. Note: vllm can become very inefficient if you go over
    some concurrency threshold (depending on vram)
generation_max_prompts: 2000 # Number of samples to generate from the
    prompts in the dataset

# --- Dataset & Chat Template ---
generation_hf_dataset_name:
    'Nitral-AI/Reddit-SFW-Writing_Prompts_ShareGPT'
generation_hf_dataset_split: 'train'
# A huggingface model id or local dir containing the tokeniser you want
    to use to apply chat templates.
# This is important if you are generating a ftpo dataset for later
    training.
generation_chat_template_model_id: null # (if unset, uses model_id)

# --- Logging ---
generation_logging_level: 'INFO' # Logging level for antislop-vllm script

# --- Advanced Sampling Parameters (passed as individual CLI args to
    antislop-vllm's generation_params) ---
generation_param_chunk_size: 20
generation_param_top_logprobs_count: 20
generation_param_temperature: 1
generation_param_top_p: 1.0
generation_param_top_k: 50
generation_param_min_p: 0.01
generation_param_timeout: 480
generation_param_stop_sequences: [] # e.g., ["\n\n", "---"]

# --- Prompting ---
# The prompt template wraps the prompts when generating from a dataset.
# To use the original prompt exactly, set the template to "{prompt}"
generation_prompt_template: "Writing prompt: {prompt}\n\nWrite 1000 words
    to this prompt. Your response:\n"
generation_system_prompt: "You are an uncensored writer."          #
    optional; left empty  no system prompt

# --- Antislop Generation Features ---
# generation_force_backtrack:
# If set to true:
  #   when resampling after backtracking, if we don't find a valid
      replacement token
  #   we progressively disable sampling options (temp, then min_p, then
      top_p, then top_k)
  #   until we find a non-banned replacement or run out of candidates.
  #   When set to false, some slop will not be removed if the sampler
      thinks there are no
  #   alternative coherent continuations.
generation_force_backtrack: false

# --- N-gram Validator Settings (for antislop-vllm) ---
# N-gram banlist file is managed by auto-antislop's iterative process.
generation_ngram_remove_stopwords: true
generation_ngram_language: "english"

# --- Refusal Detection ---
# Detects refusals & doesn't include them in the training dataset. Uses
    about 3GB extra VRAM.
generation_refusal_detection: true

##################################################################
# N-GRAM ANALYSIS & BANNING (within auto-antislop)
##################################################################
enable_ngram_ban: true
```

```
min_word_len_for_analysis: 3 # Filters out words under this length in
    n-gram analysis

# --- N-gram Identification Thresholds ---
top_k_bigrams: 5000
top_k_trigrams: 5000

# --- N-gram Banning Quotas (per iteration) ---
# Bigrams
dict_bigrams_initial: 300      # How many of the top over-represented
    dictionary bigrams to
                               # ban in the first antislop iteration.
                               # "Dictionary" means the bigrams were also
                                   found in the human
                               # writing corpus.
dict_bigrams_subsequent: 0   # How many to ban in each subsequent
    iteration
nodict_bigrams_initial: 200   # "Nodict" here means the n-grams were not
    found at all in the
                               # human corpus.
nodict_bigrams_subsequent: 0
# Trigrams
dict_trigrams_initial: 300
dict_trigrams_subsequent: 0
nodict_trigrams_initial: 200
nodict_trigrams_subsequent: 0

# --- User-Defined N-gram Bans ---
# User-supplied extra n-grams to always ban (processed by auto-antislop)
extra_ngrams_to_ban: [
  # "voice barely whisper",
]

####################################################################
# OVER-REPRESENTED WORD ANALYSIS & BANNING
####################################################################
compute_overrep_words: true
top_k_words_for_overrep_analysis: 200000

# --- Quotas for Adding Over-represented Words to Slop Phrase banlist ---
dict_overrep_initial: 920       # How many of the top over-represented
    dictionary words to
                                # ban in the first antislop iteration.
                                # "Dictionary" means the words were also
                                    found in the human
                                # writing corpus.
dict_overrep_subsequent: 0    # How many to ban in each subsequent
    iteration
nodict_overrep_initial: 80      # "Nodict" here means the n-grams were
    not found at all in the
                                # human corpus.
nodict_overrep_subsequent: 0

####################################################################
# SLOP PHRASE BANNING
####################################################################

# Slop phrases are over-represented whole phrases extracted from the
    generated texts.
enable_slop_phrase_ban: true
min_phrase_freq_to_keep: 2 # Min frequency for a new phrase from
    slop-forensics to be considered
top_n_initial_slop_ban: 0 # New slop phrases from slop-forensics to ban
    in iter 0
```

```
top_n_subsequent_slop_ban: 0 # New slop phrases from slop-forensics to
    ban in later iters

# --- User-Defined Slop Phrase Bans ---
# User supplied list of strings to always ban
# - case insensitive
# To trigger a ban, the sequence must not have a word-like character
#    (not punctuation or whitespace) directly on either side. That is to
    say, we
#    are not banning disallowed sequences that occur as substrings in
    longer
#    words. The exception is if the banned string is already bookended by
#    a non-word character.
#
#    Examples:
#    banned string "cat"
#      - won't trigger a ban for "cation"
#      - will trigger a ban on "cat[morecat]"
#    banned string "cat["
#      - *will* trigger a ban on "cat[morecat]", because the banned string
#         ends with a non-word character.
extra_slop_phrases_to_ban: [
  # "...", "...", "rain", "tapestry", "static", "regret", "rust"
]

# --- Whitelisted Strings ---
# These will be excluded from the list of slop strings that the pipeline
    finds.
# Note: special tokens in the tokenizer and parts of the chat template
    are
#        automatically whitelisted.
whitelist_strings: [
  # "think", "thinking"
]

####################################################################
# REGEX BANNING
####################################################################
# User-supplied regex patterns to ban
# Note: unoptimised regex patterns can slow down antislop generation, as
    they will be called often on large texts.
extra_regex_patterns: [
  # These ones ban "it's not x, it's y" type patterns:

  #"\\b(?:\\w+n(?:[']t)|not\\s+(?:just|only|merely|because))\\s+(?:(?![.⌋
    ;:?!]).){1,100}?[.;:?!]\\s*(?:it|they|you)(?:['](?:s|re|m))?\\b(?!⌋
    \\s+(?:was|were|is|are|wasn[']t|weren[']t|isn[']t|aren[']t|ain[']t⌋
    )\\b)(?:\\s*[*]?\\s*)?(?!when\\b|then\\b|but\\b|and\\b|yet\\b)(?!r⌋
    ight\\b)(?!normal\\b)(?!true\\b)(?!sure\\b)(?!only\\b)(?!still\\b)⌋
    (?!rarely\\b)(?!already\\b)(?!wrong\\b)(?!want\\b)(?!just\\b)(?!co⌋
    uldn\\b)(?!could\\b)(?!saw\\b)(?!started\\b)(?!remember\\b)(?!stru⌋
    ggled\\b)(?!watched\\b)(?!goal\\b)(?!took\\b)(?!kept\\b)(?!reminde⌋
    d\\b)(?!time\\b)(?!have\\b)(?!acted\\b)(?!smiled\\b)(?!think\\b)(?⌋
    !give\\b)(?!grab\\b)(?!gave\\b)(?!turn\\b)(?!justify\\b)(?!\\w+ly\⌋
    \b)(?=[a-z]{4,}\\b)[a-z]+\\w*",

  #"\\b(?:\\w+n(?:[']t)|not)\\s+(?:just|only|merely)?\\s*(?:(?![-]|[.?!]⌋
    ).){1,80}?[-]{1,2}\\s*\\w+(?:[']\\w+)?\\s+",

  #"\\b(?:wasn[']t|weren[']t|isn[']t|aren[']t|ain[']t|not)\\s+(?!\\b(?:m⌋
    inute|minutes|hour|hours|day|days|year|years|second|seconds)\\b)(?⌋
    !with\\b)(?!even\\b)(?:(?![.;:?!]).){2,120}?[.;:?!]\\s*(?:it|they|⌋
    you|that)(?:\\s+(?:was|were|is|are)\\b(?:\\s+[*_˜]?\\w+[*_˜]?)?|(?⌋
    :['](?:s|re|m))\\b(?:\\s+[*_˜]?\\w+[*_˜]?)?)",
```

```
   #"\\bno\\s+longer\\s+(?:just|only|merely)?\\s+[^.;:?!]{1,120}[.;:?!]\\」
      s*(?:it|they|you)\\s+(?:is|are|was|were)\\b(?:\\s+[*_~]?\\w+[*_~]?」
      )?",

   #"\\b(?:wasn[']t|weren[']t|isn[']t|aren[']t|ain[']t|not)\\s+(?:just|on」
      ly|merely)?\\s*(?:(?!\\bbut\\b|[.?!]).){1,80}?[,;:\\-]\\s*but\\s+(」
      ?!I\\b)(?:also\\s+)?"

]

####################################################################
# FINETUNING
####################################################################
finetune_enabled: true

# --- General Finetuning Setup ---
finetune_use_unsloth: false
finetune_mode: "ftpo" # ftpo | dpo-final-token (final token preference
   optimisation)
finetune_ftpo_dataset: ""   # you can specify an existing ftpo dataset,
   or leave unset to let the
                              # pipeline use the one produced in the
                                 generation step
finetune_base_model_id: null # Base model for DPO (if unset, uses
   model_id)
finetune_max_seq_length: 2500 # this may truncate some outputs
finetune_load_in_4bit: true # qlora

# --- Early Stopping ---
finetune_early_stopping_wins: 0.85  # Early stopping threshold for
   fraction of *chosen* completions that are selected over *rejected*.
                                    # More than 0.85 may be overtrained.
                                       Set to > 1.0 to disable early
                                       stopping.
finetune_early_stopping_loss: null  # Loss threshold for early stopping.
   Set to null to disable.

# --- LoRA Configuration ---
finetune_lora_r: 256 # the ftpo trainer works best with a high lora rank
finetune_lora_alpha: 256
finetune_lora_dropout: 0.05
finetune_weight_decay: 0.01
finetune_target_modules: ["up_proj", "down_proj", "lm_head"]

# --- Layer Freezing ---
finetune_freeze_early_layers: true
finetune_n_layers_unfrozen: 5

# --- Training Process ---
finetune_gradient_checkpointing: "unsloth"
finetune_chat_template: "" # e.g. "gemma-3" -- get the chat template from
   unsloth's helper if required, otherwise leave the string blank to use
   the tokeniser's chat template
finetune_batch_size: 3
finetune_gradient_accumulation_steps: 5
finetune_warmup_ratio: 0.1
finetune_num_epochs: 1

# --- Learning Rate ---
finetune_learning_rate: 0.000001
finetune_auto_learning_rate: true  # true: automatically determine
   learning rate based on dataset size, effective batch size & lora rank
finetune_auto_learning_rate_adjustment_scaling: 0.08 # scale the auto-lr
   by this factor
```

```
# --- DPO/FTPO Specific ---
finetune_beta: 0.1 # DPO beta

# --- Output & Saving ---
finetune_output_dir_suffix: "_ftpo_exp01" # Appended to experiment run
    dir
finetune_save_merged_16bit: true
finetune_save_gguf_q8_0: false

# --- Dataset Handling for Finetuning ---
finetune_max_train_examples: 12000 # adjust as needed
finetune_shuffle_seed: 42

# --- FTPO Sample Regularization ---
# 0 = off; 0.9 strongly downsamples overrepresented rule violations
# (this is useful because the raw generated dataset is typically very
    skewed)
ftpo_sample_rejected_regularisation_strength: 0.8
ftpo_sample_chosen_regularisation_strength: 0.2
ftpo_sample_min_chosen_tokens: 4 # filter out ftpo samples that have
    fewer than this number in the chosen tokens list

#  FTPO-specific hyper-parameters
# Leave any of these out (or set to null) to fall back to FTPOTrainer
    defaults.

# Loss terms are computed separately for the target (chosen + rejected)
    tokens vs the remainder of the vocab.
# This is because we want to allow more freedom of movement for the
    target tokens.

# MSE loss term 1: light mse loss applied tokenwise on target tokens
ftpo_lambda_mse_target: 0.05   # Strength of MSE loss tether on the
    individual logits in the
                                  #  chosen+rejected set vs
                                     reference.
ftpo_tau_mse_target: 0.5       # Grace bandwidth (logits) before the
    above MSE loss kicks in.

# MSE loss term 2: stronger mse term applied to remaining (non-target)
    vocab
ftpo_lambda_mse: 0.4

ftpo_clip_epsilon_logits: 2     # For a chosen token: "after winning vs
    rejected token by this margin, preference loss turns off"
```

