# OpenReview forum: "Antislop: A Comprehensive Framework for Identifying and Eliminating Repetitive Patterns in Language Models"
_ICLR.cc/2026/Conference — ICLR 2026 Poster_

### Official Review · Reviewer_VzHt · 2025-10-19

**Soundness:** 3
**Presentation:** 3
**Contribution:** 2
**Rating:** 6
**Confidence:** 3

**Summary:**

The paper presents Antislop, a system for reducing over-represented lexical and stylistic patterns in LLM outputs. It has three components: (1) an inference-time Antislop Sampler that detects matched strings or regex patterns and backtracks to resample with a tunable penalty, (2) an automated “slop fingerprinting” pipeline that builds pattern lists by comparing model outputs to human baselines, and (3) a training objective FTPO that localizes updates to the token where a banned pattern would begin. Experiments on creative writing tasks and standard benchmarks report large reductions in targeted patterns with small quality changes. Throughput costs of the sampler are measured, and FTPO is proposed to retain suppression without runtime overhead.

**Strengths:**

1. Improving the humanness of LLM generation is an important problem with a lot of real-world applications. This paper provides a complete recipe from decoding to training, including a backtracking sampler, an automated fingerprinting step, and a localized training objective.
1. The decoding overhead is measured and a training procedure is designed to internalize the discouragement of undesired patterns.
1. Experiments span multiple model sizes and families and include both creative writing outcomes and collateral checks on standard tasks, with ablations and throughput analyses that support the claims

**Weaknesses:**

1. Training to discourage specific tokens or sequences has been studied somewhat extensively [1]. The contribution here can feel incremental without a stronger theoretical or empirical separation from this line of work.
1. Broader testing on recent creative-writing setups would strengthen the work. For instance, testing on poems, stories, and jokes could serve as a complementary setting [2].
1. Does suppressing the most prominent patterns surface other generation patterns? These patterns may not be human like either. I am curious about the results of the study in section 3 applied to the fine-tuned model.

[1] Neural Text Generation with Unlikelihood Training. Welleck, Kulikov, Roller, Dinan, Cho, Weston. 2019.  ￼
[2] Verbalized Sampling: How to Mitigate Mode Collapse and Unlock LLM Diversity. Zhang et al. 2025.

**Questions:**

Please refer to the weaknesses.

---

> ### Author Response · Authors · 2025-11-28
> **Response to VzHt**
>
> We thank the reviewer for the thorough evaluation and for recognizing our "complete recipe" from decoding to training. We have addressed your specific concerns regarding the risk of "new slop", the distinction from Unlikelihood Training, and broader evaluation settings.
>
> **1\. "New Slop" Emergence:** This is a great question: does suppressing specific patterns simply cause the model to surface *other* non-human patterns (the "whack-a-mole" effect)? We conducted a rigorous new analysis (**Appendix M**) to test this:
>
> * **Independent Fingerprinting:** We computed the top-40 most over-represented words for the FTPO-finetuned model *independently* of the training banlist. The average over-use ratio vs human writing dropped from **1,439x (baseline Gemma-3-27B)** to **394x (FTPO finetuned)**, a **73% reduction**. Similarly, we observed a **36% reduction** in over-represented trigrams. This proves the total volume of "slop" has been reduced, not just shifted to new patterns.
> * **Entropy & Distribution Flattening:** We analyzed the log-probability distribution of the top-20 tokens at the exact positions where banned tokens would have occurred (across 400 samples). Entropy increased from **1.34 (baseline)** to **1.93 (FTPO)**, indicating a flatter distribution where the model considers more diverse alternatives. Importantly, this improvement is broad: entropy even at random non-banned positions increased from **0.59 to 0.89**.
>
> These results confirm that the model is not substituting "Cliché A" with "Cliché B"; instead, it is genuinely dispersing probability mass across a diverse set of context-appropriate alternatives.
>
> **2\. Distinction from Unlikelihood Training.** The unlikelihood loss of Welleck et al. (2019) is a general mechanism: it penalizes log-probability assigned to tokens or sequences that are tagged as “negative,” but leaves three design choices open: (1) how negatives are identified in context, (2) what positive objective they are combined with, and (3) what additional regularization, if any, constrains the rest of the distribution. Our FTPO pipeline fixes these choices in a very specific way for slop:
>
> * **Where to apply the negative signal.** We only apply it at final tokens that initiate empirically over-represented patterns, identified by comparing model outputs against a human corpus; the rest of the sequence is untouched.
> * **How to define the positive side.** We do not rely on generic MLE; instead, we construct final-token preference pairs where the rejected token is the start of a slop pattern and the chosen tokens are coherent high-probability alternatives from the same position.
> * **How to regularize the distribution.** We work in logit space, adding separate MSE tethers for target (chosen+rejected) and non-target logits, with a dead-zone window and a margin-based switch-off so that once chosen tokens win by a margin, gradients turn off.
>
> In that sense, FTPO does not replace unlikelihood as a concept; it instantiates something conceptually close in the preference term of the FTPO loss function, where rejected tokens are punished. We additionally make several specific implementation choices (final-token preference pair structure, two-part logit-space regularization, win margin) tailored to suppressing over-represented stylistic patterns with minimal collateral change to the rest of the vocabulary.
>
> **3\. Broader Testing:** Addressing your request to evaluate beyond our primary Nitral (Reddit) and Longform Writing benchmarks, we extended our testing to include an **Essays and Creative Writing dataset**, which interleaves prompts from analytical and creative writing styles. We trained a single FTPO model on this mixed data to assess generalization. The results confirm robustness across domains: the model successfully suppressed slop in both contexts (reducing hits from 32.27 to 5.61 per 1k words) while actually **improving longform writing quality scores** (54.7 $\\to$ 57.4). This demonstrates that FTPO can handle diverse stylistic constraints without degradation. We have incorporated these cross-domain results into **Section 6.2** of the revised manuscript.
>
> —----------
>
> We thank you for your insightful review, which drove these improvements. We have dedicated significant time and effort to these new experiments, particularly the independent entropy analysis and cross-domain validation, to ensure your concerns are fully resolved. We believe these revisions substantially strengthen the manuscript and respectfully ask you to consider raising your score. If you have any further questions or suggestions for additional validation, please let us know.

---

### Official Review · Reviewer_DMCK · 2025-10-29

**Soundness:** 4
**Presentation:** 3
**Contribution:** 3
**Rating:** 6
**Confidence:** 3

**Summary:**

This paper introduces ANTISLOP, a framework designed to identify and mitigate repetitive, low-diversity “slop” patterns in large language model outputs. The authors propose Final Token Preference Optimization (FTPO), a novel preference optimization method that regularizes model logits through a margin-based loss to discourage redundant phrasing. Combined with the Antislop Sampler, which automatically generates preference pairs from repetitive outputs, the framework enables self-supervised fine-tuning to improve linguistic diversity and stylistic quality. Extensive experiments across creative writing and reasoning datasets demonstrate that ANTISLOP effectively reduces repetition while maintaining fluency and coherence.

**Strengths:**

"slop" is a very interesting entry point. The experiments are comprehensive and well-executed, covering multiple datasets and comparing FTPO with existing training methods.

**Weaknesses:**

* The paper lacks qualitative examples illustrating how FTPO replaces repetitive phrases with new ones. Does the model exhibit “over-pruning” or semantic drift? Without such examples, the linguistic significance of the contribution remains unclear.

* The theoretical foundation of FTPO’s uniqueness and generalization is underdeveloped. At present, it still feels quite similar to DPO. You could first address this from a writing and framing perspective — for instance, by clarifying why existing preference optimization methods fail to handle “slop” effectively. Highlighting this gap might help crystallize the underlying theoretical novelty of FTPO.

**Questions:**

Why you choose mmlu and gsm8k? In my opinion, they are more related to scientific writing (maybe), rather than creative writing. But your dataset is based on creative writing.

---

> ### Author Response · Authors · 2025-11-28
> **Response to DMCK (Part 1)**
>
> We thank the reviewer for recognizing the value of the "slop" framing and the comprehensive nature of our experiments. We have specifically addressed your requests for qualitative evidence and theoretical sharpening through significant revisions.
>
> **1\. Semantic Drift:** You noted that our metrics might hide "over-pruning" or semantic drift. We have addressed this in two ways:
>
> * **Quantitative Verification:** To rigorously verify the absence of drift, we conducted a new **embedding analysis** (Appendix N). We found that the cosine distance between the baseline and FTPO models is minimal ($0.098$) when compared to the cosine distance between the baseline model’s own outputs on a given prompt under temperature sampling ($0.079$). This represents a **5.3x smaller** shift than we measured when injecting a style prompt ("Write in the style of Hunter S. Thompson"). Furthermore, token entropy at suppression sites increased (**1.34 $\\to$ 1.93**), confirming the model is expanding its vocabulary rather than collapsing.
> * **Contextual Awareness (Appendix A):** We demonstrate **soft-banning** in action with a parameter sweep on ban-strength. The model suppresses "tapestry" when it would otherwise appear as an over-used metaphor, but correctly allows it when the prompt explicitly asks for an essay about textile history.
>
> **2\. Theoretical Novelty: FTPO vs. DPO**
>
> You correctly noted that the theoretical distinction between FTPO and DPO was underdeveloped. We outline in Section 5.3 why standard preference optimization fails at "negative constraints" (slop suppression) and how FTPO solves this structurally:
>
> * **The Failure Mode of DPO (Probability-Space Coupling):** DPO operates in probability space using KL divergence ($D\_{KL}(\\pi\_\\theta || \\pi\_{ref})$). Due to the softmax normalization constraint ($\\sum p\_i \= 1$), strongly suppressing a rejected "slop" token *mechanically* forces that probability mass to be redistributed across the remaining vocabulary. DPO controls this with a single $\\beta$ parameter, making it impossible to treat target tokens differently from the rest of the vocabulary. This creates **"global collateral damage,"** leading to the **lexical diversity collapse** we observe in Table 2 (DPO diversity drops to \~74% of baseline at 8k patterns).
> * **The FTPO Solution (Logit-Space Decoupling):** FTPO operates in logit space using MSE loss. Because we are not working with normalized distributions, we can lower the logit of a target token without mechanically forcing compensatory increases elsewhere. We implement a **two-part regularization** impossible in DPO:
>   1. **Target Tokens:** Light MSE regularization with a "grace window" ($\\tau$), allowing larger movements.
>   2. **Non-Target Tokens:** Strong MSE anchoring ($\\lambda\_{nontarget}$), keeping the broader vocabulary tightly tethered to the reference model.
> * **Stability via Gradient Deactivation:** DPO lacks a mechanism to turn off training signal on already winning pairs; chosen & rejected logits continue to diverge during training, only bounded by DPO’s sigmoid function. In contrast, FTPO implements both a reference tether and a **win-margin gradient deactivation** via the weight term $w\_c \= \\text{clamp}((m \- \\Delta\_c)/m, 0, 1)$. Once chosen tokens win by margin $m$, gradients automatically zero out. This prevents overtraining, allowing FTPO logits to plateau stably (\~3 logits from reference) and maintaining **97.7% lexical diversity**.

---

> > ### Author Response · Authors · 2025-11-28
> > **Response to DMCK (Part 2)**
> >
> > **3\. Evaluation Strategy (Question 1\)** You asked why we employ MMLU and GSM8K in a creative writing study. This is a deliberate design choice to measure two distinct axes of model performance: **Optimization Success** (in the target domain) versus **Capability Preservation** (in the general domain).
> >
> > * **Sentinel Tasks for Catastrophic Forgetting:** MMLU and GSM8K serve as "sentinel tasks." When fine-tuning a model to suppress specific stylistic patterns, there is a risk of damaging the model's fundamental reasoning and knowledge capabilities (catastrophic forgetting). These benchmarks allow us to verify that our intervention improves style without "lobotomizing" the model's core intelligence.
> > * **The "Complete Story":** Our results demonstrate this dual success: FTPO achieves **90% slop reduction** and improved creative writing quality (Target Success), while simultaneously maintaining **98–99%** of baseline performance on reasoning tasks (Capability Preservation).
> > * **Contrast with Baselines:** If we only evaluated writing, we could not distinguish genuine improvement from damaging overfitting. Indeed, DPO fails this safety check, degrading reasoning metrics by **2–5%**, whereas FTPO preserves them.
> >
> > —--------------
> >
> > Thank you for the constructive feedback. We have added substantial new content including qualitative examples, theoretical clarification, and evaluation rationale. We believe these additions significantly strengthen the paper's clarity and impact, and we hope you'll find them sufficient to support acceptance and raise your score. We're happy to provide additional examples or clarifications if needed.

---

### Official Review · Reviewer_HpSi · 2025-11-01

**Soundness:** 2
**Presentation:** 2
**Contribution:** 2
**Rating:** 2
**Confidence:** 4

**Summary:**

This paper proposes a new method named Antislop to prevent the large language model (LLM) from generating repetitive or overused sequences while maintaining its generalization capability. More specifically, Antislop can build a preference dataset by identifying the overused patterns in the target LLMs and then regenerating the sequences from the starting point of the detected pattern with resampling. Based on the preference dataset, the finetuning method called final token preference optimization (FTPO) is applied to adapt the target LLM, such that it could generate repetitive sequences much less frequently. Experimental results show that the FTPO can achieve a higher suppression rate of overused patterns than DPO and logit-based token banning while maintaining its performance on the downstream tasks.

**Strengths:**

The strengths of the paper are listed as follows.
1. It is great that the authors have a plan to release their dataset and codes for reproducibility.
2. The paper is well motivated. Generating repetitive patterns is a big issue for LLMs and could affect the output quality significantly. Therefore, an effective method to address this issue is important.

**Weaknesses:**

The weaknesses of the paper are listed as follows.
1. The training-based method is model-specific. It requires building a different preference dataset for different models, which is costly and thus limits its application and impact.
2. The experimental results are not convincing enough. First, the authors only compare their method with DPO and logit-based token banning. There are many existing works mentioned in the related work section. However, the authors did not provide any justification for not comparing with the remaining existing works on this topic. Second, the authors did not provide an ablation study of the regularization terms defined in Section 5.2.
3. Some important technical details are not clear. I list some of them as follows.
a. When quantifying the slop, it is unclear how to define the pattern $p$.
b. When detecting the banned patterns, Antislop will lower the initiating token’s probability. However, it is unclear how to adjust the probabilities of the remaining tokens.
c. In FTPO, it is unclear how to obtain the set of chosen tokens for each rejected token.
d. Many hyperparameters in FTPO are logit-space operations. Will it make the choice of hyperparameters tricky since the search space is large?
e. In equation of $L_{target}$, it is unclear how to decide the value of $y_{ref}$.
4. The writing and organization of the paper can be improved for better readability. First, it would be better to place Figure 1 on page 5 since the first time it is referred to is on page 5. Second, it would be better if the authors could provide a figure to illustrate the difference between the proposed method and the other types of existing works to highlight the technical novelty of the proposed method. Third, before you describe the loss function of FTPO, it would be better to clearly define the form of the dataset. For example, does each data sample have labels or pseudo-labels? How to get these labels? Is the loss only calculated on the response part of each sample? Last, when introducing the benchmarks, it would be better to describe the evaluated dataset and the metrics separately instead of merging them into the same part.
5. The FTPO can prevent the target LLMs from generating the overuse patterns detected by the preference dataset. However, it is unclear whether it will incur new overuse patterns in the finetuned LLMs.
6. It would be better if the authors could provide the cost analysis of the FTPO, which is important to show the scalability of the proposed method, especially when the target LLM is large.

**Questions:**

Please refer to the weaknesses section for details.

---

> ### Author Response · Authors · 2025-11-28
> **Response to HpSi (Part 1)**
>
> We thank the reviewer for their detailed assessment. We appreciate the acknowledgment that the problem is well-motivated and important. We have addressed your concerns regarding cost, baselines, and technical definitions through significant revisions and new experiments.
>
> **1\. Scalability, Cost, and Model Specificity:** You raised a concern that building model-specific datasets limits applicability (W1) and requested a cost analysis (W6). We have added a comprehensive breakdown (**Appendix C**) and a cross-model analysis (**Appendix L**) to demonstrate that the method is highly scalable, economically viable, and reusable:
>
> * **Negligible Cost & Energy:** The complete pipeline for `gemma-3-12b`, generating 2,000 samples and performing FTPO fine-tuning, requires only **\~3.8 hours on a single H100 GPU** (1.5h for generation \+ 2.3h for training). The total cloud cost is approximately **$7.50**, with an energy consumption of **\~2.6 kWh**. This is orders of magnitude cheaper than standard pre-training or RLHF workflows.
> * **Reusability:** We analyzed slop fingerprint stability, finding a **56% overlap** within model families (12B vs 27B) and **\~31% overlap** across families (e.g., Gemma vs Mistral). This indicates that datasets generated for one size can be re-used with reasonable coverage for other models.
> * **Scalability:** FTPO scales efficiently to larger models because it operates on final-token preference pairs rather than full-sequence generation. Training the 12B model requires 42 GB VRAM (fitting on a single A100/H100), while the 27B model requires 68 GB (using QLoRA), confirming the method is viable for large-scale production models.
>
> These results confirm that treating slop as a "model-specific" pathology to be cured via a targeted fine-tune is a practical and low-cost strategy for production deployment, rather than a limitation.
>
> **2\. Baseline Comparisons:** You noted the limited comparison to DPO/Token Banning. We have expanded our baselines significantly:
>
> * **Inference-Time Baselines:** We added comparisons to modern decoding methods like **XTC** and **DRY** (Appendix O), showing they fail to meaningfully suppress slop. We also compare against **Token Banning**, which achieves only **61% suppression** (at 2k patterns) while severely degrading writing quality, confirming that hard constraints are insufficient compared to Antislop's \~100% suppression.
> * **Training Baselines:** We prioritize DPO as the baseline comparison, as the most widely used offline preference optimization algorithm. We clarify in the text that Unlikelihood training defines only the loss component punishing unwanted sequences/tokens; whereas FTPO defines the full loss objective and the preference pair specification. We are currently completing additional experiments with **KTO** and will include full results in the final manuscript.
>
> **3\. Ablation Studies:** Addressing your request for justification of the regularization terms, we provide a comprehensive ablation study (Appendix E) validating the necessity of both critical FTPO hyperparameters:
>
> * **Margin Clipping (clip\_epsilon):** Disabling this (setting to 16\) causes catastrophic collapse to single-word repetitions (**Quality drops to 19.57/100**). Our default (2.0) achieves the optimal balance (**Quality 67.89, vs. 67.80 baseline**).
> * **Target Tether (lambda\_mse\_target):** Disabling this (0.0) enables fast but degraded training (**Quality 39.65/100**). Over-regularizing (0.4) preserves quality but impairs learning (**Suppression only 55.86%**).
>
> These results confirm that our default settings ($0.05$) balance quality and suppression optimally and that the "soft-touch" components are critical for stability.

---

> > ### Author Response · Authors · 2025-11-28
> > **Response to HpSi (Part 2)**
> >
> > **4\. "New Slop" Emergence:** To address the risk of substituting new overuse patterns (the "whack-a-mole" effect), we re-computed the "slop fingerprint" for the FTPO model – its top 40 most over-represented words & trigrams vs human baseline – *independently* of the training banlist. We found a **73% reduction** in the average usage ratio vs human baseline for the top 40 words, and a **36% reduction** for trigrams. Crucially, entropy at suppression sites increased by **44%** ($1.34 \\to 1.93$), and even at random positions ($0.59 \\to 0.89$), indicating broader distributional improvements. Finally, we conduct a new embedding analysis experiment, detailed in Appendix N. Our results show a **10.1% increase** in intra-model diversity, while semantic drift is minimal, **5.3x smaller** than the shift caused by injecting a style prompt ("Write in the voice of Hunter S. Thompson,"). This confirms FTPO suppresses slop without shifting to new repetitive modes, and without excessive semantic drift.
> >
> > 5**. Technical Clarifications:** We have inserted the following explicit clarifications into the main text (Sections 4.1 and 5.2) to ensure reproducibility:
> >
> > * **(a) Pattern Definition:** We define a pattern $p$ as any n-gram where $\\rho(p) \= f\_{LLM}(p) / f\_{human}(p)$ exceeds a fixed threshold (10x for words, 3x for trigrams). We extract these by generating 2,000 outputs, removing stop-words, and comparing frequencies against human baselines (Wordfreq \+ Reddit/Gutenberg corpora).
> > * **(b) Probability Adjustment:** When Antislop lowers the initiating token's probability ($p'\_{banned} \= p\_{banned} \\cdot 10^{-10s}$), we simply **renormalize** the entire distribution ($p'\_i \= p\_i / \\Sigma\_j p'\_j$). This preserves the relative ordering of non-banned tokens while ensuring $\\sum p'\_i \= 1$, followed by min-p filtering (threshold 0.1).
> > * **(c) Chosen Tokens:** At each backtracking event, we cache the top-$k$ logits ($k=20$), remove the rejected token, renormalize, and sample 4–8 viable alternatives. These form the "Chosen" set for FTPO.
> > * **(d) Hyperparameters:** We found our default values transfer robustly across Gemma, Llama, Mistral, and GLM families because the loss uses relative scales.
> > * **(e) Reference Logits ($y\_{ref}$):** We clarified that $y\_{ref}$ represents the logit values from the frozen base model. To avoid redundant forward passes, we cache these logits during the data generation phase.
> >
> > **6\. Writing and Organization:** We have implemented all your suggestions: moving **Figure 1** to page 5, formally defining the **Dataset Format**, and separating **Datasets** and **Metrics** in Section 6.1.
> >
> > \---
> >
> > We thank you again for your constructive review, which drove these improvements. We have dedicated significant time and effort to these rigorous new experiments and analyses, resulting in a substantially stronger manuscript. We believe these comprehensive additions firmly address the concerns regarding novelty, scope, and stability, and we respectfully ask you to reconsider your score. If you have any further questions or require additional experiments to resolve outstanding concerns, please let us know, we are happy to address them.

---

### Official Review · Reviewer_wHcb · 2025-11-01

**Soundness:** 3
**Presentation:** 3
**Contribution:** 3
**Rating:** 6
**Confidence:** 4

**Summary:**

The paper proposes an approach to discourage repetitive patterns generated from LLMs. It first gives an approach to identify repetitive pattern by sampling model outputs and comparing frequencies of ngrams with standard frequencies. Once identified they propose and compare a couple of ways to reduce their occurrence while maintaining generation quality. The first approach is to backtrack whenever a bad pattern is detected and then regenerate from that point by sampling again while discouraging the token that led to the bad pattern by renormalizing probabilities. To mitigate the decrease in throughput due to the backtracking, they propose to use finetuning approaches using synthetic data. They propose a new approach FTPO which precisely edits logits for preferred and rejected tokens, and is able to maintain the generation quality better than DPO finetuning while reducing the undesired n-gram frequency more substantially.

**Strengths:**

- The problem of reducing repetitive patterns in LLM generation is well motivated and an important one to solve in order to get natural texts from LLMs.

- The soft-banning approach proposed provides customizable adjustment strengths which are useful to add flexibility.

- FTPO approach is well motivated with well thought out components to encourage the logit distribution you want while constraining the logits so that they don’t change too much, with different components of the loss encouraging different goals.

- The results show that FTPO can provide better tradeoff between quality of output and reduction in slop than DPO and other approaches (Figure 3).

**Weaknesses:**

- The slop fingerprinting would be dependent on what queries are fed into the model to get the outputs, and so it might be difficult to get fingerprints (overrepresented n-grams) that are more universal (i.e. show up in wide-scale use in general domains).

- Not enough attention is paid to detecting and filtering semantic fingerprints that are beyond specific ngrams and of more semantic nature. For example patterns like ”it’s not just X, it’s Y.” are discussed in the paper very briefly in section 6.5 which proposes using regular expressions to detect them, but it might lead to reduction in output quality if we prevent entire clause structures like that from being generated (which is not studied in the work).

- The approach of soft banning would require tuning strength parameters to use for renormalizing probabilities and it is not clear to me that a single parameter setting will work well across multiple domains in a wide scale real-life deployment.

**Questions:**

- What was the value of n in your ngrams for the investigation in section 3 and how did you decide on a value for that?

- I would suggest adding equation numbers to equations (e.g. the loss equations in FTPO) for ease of referring to them

- In line 323, it says "we freeze all layers except the last 5 and lm head" This does not sound like standard practice.. do you do experiments with training all layers via LoRA and see how that works compared to only the last 5 layers? It is okay to maybe use a lower rank LoRA but use it on all layers if gpu memory constraints are there.

---

> ### Author Response · Authors · 2025-11-28
> **Response to wHcb (Part 1)**
>
> We thank the reviewer for the positive assessment of our soft-banning approach and FTPO design. We appreciate your recognition of the method’s flexibility. We have addressed your valid points regarding domain specificity, semantic patterns, and technical hyperparameters through new experiments and clarifications.
>
> **1\. Universality of Slop Fingerprints:** You correctly noted that fingerprints are inherently domain-specific. To rigorously test the method's applicability across domains, we conducted **New Cross-Domain Experiments**:
>
> * **Method:** We generated a preference dataset using a mixture of **Creative Writing** and **Analytical Essay** prompts, then trained `gemma-3-27b` on this mixed dataset with FTPO.
> * **Suppression Results:** The finetuned model successfully suppressed slop in *both* domains. Slop list hits dropped from **32.27** (baseline) to **5.61** (finetuned) per 1k words – an **83% suppression rate** of the cross-domain banlist.
> * **Quality Results:** Crucially, writing quality was not degraded; the finetuned model actually scored **57.4** on the longform writing rubric compared to the baseline of **54.7**.
>
> These new results, which we have added to Section 6.2, demonstrate that while specific "slop" phrases vary by domain, the FTPO pipeline effectively generalizes across them when trained on mixed data.
>
> **2\. Semantic Patterns Beyond N-grams:** You raised a crucial point about non-lexical patterns (e.g., *"It's not X, it's Y"*). We address this on two levels:
>
> * **Regex Experiments (Section 6.5):** We successfully applied Regex Bans to the structural template *"It's not \[^,\]+, it's \[^.\]+"*. Baseline `qwen3-4b` produces **1.10** of these patterns per 1,000 characters, while our regex-ban version produces **zero**, with no degradation in writing quality.
> * **The Challenge of Complex Semantic Slop:** We acknowledge that "slop" is a multi-dimensional phenomenon that extends beyond n-grams to include **structural clichés** (sentence templates), **conceptual tropes** (e.g., "shimmering" magic systems), and **narrative repetition** (e.g., specific character archetypes like "Elara"). Capturing these higher-level dimensions is a known open problem in NLP for several reasons:
>   1. **Semantic Ambiguity:** Distinguishing between a "trope" and a "genre convention" requires deep semantic understanding rather than surface-level matching.
>   2. **Computational Cost:** While solutions exist, such as **embedding-based clustering** to find conceptual repetition or **syntactic parse trees** to identify structural overuse, they are computationally prohibitive to run inside the training loop compared to our lightweight n-gram/regex checks.
>   3. **Subjectivity:** Defining "thematic slop" often requires an **LLM-as-judge** or human heuristic, which is harder to standardize than statistical over-representation.
> * Our current framework deliberately focuses on the lexical and regex-definable structural layers because they allow for **tractable, high-throughput training** while still capturing the vast majority of immediately recognizable "AI-ese."
>
> **3\. Single Ban Strength Parameter:**
>
> You raised a valid point about the limitations of a single global parameter ($s$). We clarify the behavior of this parameter empirically:
>
> * **Empirical Validation (Appendix A):** We provide a full parameter sweep of $s$. This analysis demonstrates the effective operating range of the parameter, showing how it allows users to smoothly trade off between suppression rigor and output probability mass without needing to tune hundreds of individual weights. We find that with $s$ set to 0.4-0.5, 90% of the banned list is suppressed, while still allowing sampling of banned words that are specifically requested by the user in 100% of test cases (e.g. “Write an essay about tapestry weaving”).
> * **Per-Pattern Granularity:** If tokenwise probability adjustments are desired, this is possible in our architecture as our sampler implementation **already supports per-pattern weights**. Power users can supply a dictionary mapping specific patterns to distinct ban strengths (e.g., applying $s=1.0$ for safety blocks but $s=0.1$ for stylistic preferences). The runtime cost to apply tokenwise adjustments is negligible.

---

> > ### Author Response · Authors · 2025-11-28
> > **Response to wHbc (Part 2)**
> >
> > **4\. Technical Questions**
> >
> > * **N-gram Values (Section 3):** We used **bigrams (n=2) and trigrams (n=3)**. We found that 4-grams and higher appear too infrequently (often \<5 occurrences across 2,000 samples) to be reliably identified as "slop". Bigrams and trigrams strike the optimal balance between capturing phrasal patterns and generalizing.
> > * **Layer Freezing:** We train only the **last 5 layers \+ head**. We ran informal ablations showing that training *all* layers with LoRA converged faster but led to **overfitting and repetition artifacts**. Freezing early layers acts as a necessary regularizer.
> > * **Equation Numbering:** We have numbered all loss equations as requested.
> >
> > \---
> >
> > We thank you again for your constructive review, which drove these improvements. We have dedicated significant time and effort to these new experiments, specifically the mixed-domain training and regex suppression analysis, to ensure your concerns are fully resolved. We believe these revisions substantially strengthen the manuscript's generalizability claims, and we respectfully ask you to consider raising your score. If you have any further questions or concerns, please let us know.

---

### Author Response · Authors · 2025-11-28
**General Reply to All Reviewers (Part 1)**

We thank the reviewers for their encouraging and constructive feedback. We are glad that the reviewers found the problem of "slop" well-motivated and our pipeline comprehensive. Your questions and critiques regarding "new slop" emergence and theoretical distinctiveness have driven us to conduct substantial new experiments that significantly strengthen the paper's core contributions.

In this revision, we introduce **major new experimental analyses**, **expanded baselines**, and **qualitative case studies**:

**1\. New Comparisons to Inference-time Methods (Appendix O)**: To compare our methods against XTC and DRY samplers, we generate 2,000 creative writing outputs and compare the average over-use ratio vs human writing of the top 40 most over-represented words and trigrams.

| Method | Top 40 Words Avg Usage Ratio vs Human Writing | Top 40 Trigrams Avg Usage Ratio vs Human Writing |
| ----- | ----- | ----- |
| Baseline Gemma3-27b | 1439x | 173x |
| FTPO-Finetuned Gemma3-27b | **394x** | **111x** |
| XTC Sampler | 1267x | 205x |
| DRY Sampler | 1442x | 168x |

XTC and DRY samplers can reduce intra-document repetition but cannot detect statistical patterns across independent generations, producing only minor differences in over-used pattern frequencies. Our FTPO finetuned model **reduces the average over-use ratio vs human writing from 1439x to 394x, a 73% reduction.**

**2\. "New Slop" Analysis (Appendix M):** To address the concern that suppressing specific patterns might simply substitute new clichés (the "whack-a-mole" effect; VzHt, HpSi), we re-calculated the top-40 over-represented patterns for the FTPO finetuned model independent of the training banlist. We observed a  **73% reduction** in the average usage ratio vs human writing for the top 40 words and **36%** reduction for trigrams. Concretely:

**Top over-represented words for gemma-3-27b (baseline):**

 1\. elara: ratio 18,861x, count 355

 2\. logline: ratio 3,739x, count 109

 3\. worldbuilding: ratio 3,121x, count 240

 4\. grimdark: ratio 2,819x, count 70

 5\. unsettlingly: ratio 2,275x, count 54

**Top over-represented words for gemma-3-27b-antislop (FTPO finetuned):**

 1\. elara: ratio 1,552x, count 39

 2\. unusualness: ratio 1,496x, count 18

 3\. lysandra: ratio 1,330x, count 35

 4\. outlandishness: ratio 1,169x, count 12

 5\. logline: ratio 1,028x, count 40

It’s expected that some words will become more prevalent after finetuning, as our loss function preferentially selects for alternate candidates to the rejected tokens. Two mechanisms prevent the alternate tokens from becoming severely over-represented “new slop”:

1. Gradients are distributed over several alternate chosen tokens, not a single replacement token for the rejected.
2. The win-margin disables gradients that are winning vs rejected by a specified amount, preventing chosen tokens from being overtly preferred.

The demonstrated effect is that the list of highest over-represented words in our fine-tuned model have collectively much lower frequency than in baseline gemma-3-27b.

**Entropy at suppression sites increased from 1.34 to 1.93**, and embedding analysis (Appendix N) shows **10.1% increase in intra-model semantic diversity**. This confirms FTPO does not simply swap old slop for new; rather it genuinely disperses probability across diverse and context-appropriate alternatives.


**3\. Generalization (Domains, Semantics & Baselines):** To address specificity concerns (**wHcb**, **HpSi**), we expanded evaluations in four ways:

* **Cross-Domain:** A single FTPO model trained on mixed **Creative Writing and Essays** suppressed slop in *both* domains (32.27 $\\to$ 5.61 hits/1k words) and **improved Longform Writing score** (54.7 $\\to$ 57.4).
* **Semantic Patterns:** We successfully applied **regex bans** to clause-level patterns (e.g., *"It's not X, it's Y"*), achieving 100% suppression without degrading quality (Section 6.5).
* **Cross-Model Reusability:** We found a **56% overlap** in slop fingerprints within model families (Gemma 12B vs 27B), confirming that training datasets can be efficiently reused.

---

> ### Author Response · Authors · 2025-11-28
> **General Reply to All Reviewers (Part 2)**
>
> **4\. Semantic Analysis on Embedding Distance (Appendix N):** To check that FTPO isn’t quietly shifting the model into a different semantic regime, we ran an embedding-based analysis over 500 creative-writing prompts. For each prompt, we sampled 10 completions from the baseline \\texttt{gemma-3-27b} and 10 from the FTPO-finetuned model, using temperature 1 and identical decoding settings. We then embedded all completions with a fixed encoder (\\texttt{google/gemini-embedding-001}) and compared cosine-distances.
>
> We looked at three quantities:
>
> * within-model diversity for each model (average pairwise cosine distance between that model’s own 10x samples per prompt),
> * the cross distance between baseline and FTPO completions for the same prompts,
> * and a reference “strong style shift” where we compared baseline outputs to baseline outputs generated with an added instruction (“Write in the voice of Hunter S. Thompson”).
>
> | Comparison | Mean cosine distance |
> | ----- | ----- |
> | Baseline Gemma3-27B (within-model) | 0.079 |
> | FTPO finetuned (within-model) | 0.087 |
> | Baseline vs FTPO | 0.098 |
> | Baseline vs baseline \+ style instruction | 0.180 |
>
> Baseline within-model distance is 0.079, and FTPO within-model distance is 0.087, which is a 10.1% increase and corresponds to slightly more varied responses to the same prompt. The baseline–FTPO distance is 0.098, only modestly larger than the baseline’s own within-model variation, and much smaller than the distance induced by the style instruction (0.18). In other words, FTPO moves the model more than temperature sampling alone, but far less than a deliberate style change. Taken together with the entropy and over-representation results, this supports the picture that FTPO is not drastically redefining the model’s overall style or capabilities; it is making targeted adjustments of over-used continuations while keeping the broader semantics and behaviour close to the original model.
>
> **5\. FTPO vs. DPO (Mechanism & Ablations):** Addressing the comparison requested by **DMCK** and **VzHt**, we clarify the structural necessity of FTPO over DPO through new theoretical framing (Section 5.3) and empirical tracking (Figure 4b). Theoretically, DPO’s probability-space constraint forces a **global redistribution of mass** when suppressing tokens, whereas FTPO’s logit-space MSE allows for **surgical suppression**. Empirically, we show DPO logits diverge unboundedly (\>7 logits), whereas FTPO stabilizes due to its win-margin gradient clipping mechanism and two-part MSE reference tether. We further detail an **ablation study** (Appendix D) confirming that removing clip\_epsilon\_logits or λ\_mse\_target leads to model collapse, validating that these components are essential for stability.
>
> **6\. Practicality & Capability Preservation:** Addressing scalability (**HpSi**) and capability preservation (**DMCK**), we added a comprehensive cost analysis (Appendix C.1). The full pipeline for a 12B model requires only **\~3.7 hours on a single H100** (\~$7.50). Furthermore, FTPO maintains **98-99%** of baseline performance on MMLU and GSM8K, whereas DPO degrades these metrics by  **2-5%**, confirming our intervention is not destructive.
>
> **7\. Technical Clarifications:** Finally, per reviewer requests (**HpSi**, **wHcb**), we have improved the manuscript organization, added equation numbers (Section 5.2), and formally defined pattern selection criteria (Section 3.1) and probability adjustment mechanisms (Section 4.1 and Appendix A).
>
> We believe these new experiments and structural revisions address the concerns regarding novelty, scope, and stability. We respectfully ask the reviewers to reconsider their scores in light of these substantial improvements.

---

### Meta-Review · Area_Chair_mxZ9 · 2026-01-10

**Summary:**

Focusing on the single negative review, the two primary / substantive concerns raised were:

- Worries about the cost of building model specific datasets.
- The authors initially only compared to two baselines.

There were some additional minor clarity and writing concerns that aren't really a factor here for decision making.

**Reviewer Concerns:**

I do have to say, the authors' response to both of what I believe are the main concerns of the sole rejecting reviewer are pretty convincing.

To the worry about building model specific datasets being extremely expensive: the authors' indicate that it amounts to about $7.50 of cloud costs.

The authors do add a handful of additional comparisons as well.

**Reviewer Scores:**

Tough to say. I think it's reasonable to say that the cost concern was very clearly addressed in the mind of any reasonable reviewer. Whether or not the additional baselines would have gotten there is challenging, although they are certainly appreciated.

---

### Decision · Program_Chairs · 2026-01-26

Accept (Poster)